# Coupled ion binding and structural transitions along the transport cycle of glutamate transporters

**Grégory Verdon[1]\*[†‡§], SeCheol Oh[1†], Ryan N Serio[2], Olga Boudker[1]\***

[1]Department of Physiology and Biophysics, Weill Cornell Medical College, New York, United States; [2]Department of Pharmacology, Weill Cornell Medical College, New York, United States

**\*For correspondence:**
g.verdon@imperial.ac.uk (GV);
olb2003@med.cornell.edu (OB)

[†]These authors contributed equally to this work

**Present address:** [‡]Department of Life Sciences, Imperial College London, South Kensington, United Kingdom; [§]Research Complex at Harwell, Rutherford Appleton Laboratory, Didcot, United Kingdom

**Competing interests:** The authors declare that no competing interests exist.

**Reviewing editor**: Benoit Roux, University of Chicago, United States

**Abstract** Membrane transporters that clear the neurotransmitter glutamate from synapses are driven by symport of sodium ions and counter-transport of a potassium ion. Previous crystal structures of a homologous archaeal sodium and aspartate symporter showed that a dedicated transport domain carries the substrate and ions across the membrane. Here, we report new crystal structures of this homologue in ligand-free and ions-only bound outward- and inward-facing conformations. We show that after ligand release, the apo transport domain adopts a compact and occluded conformation that can traverse the membrane, completing the transport cycle. Sodium binding primes the transport domain to accept its substrate and triggers extracellular gate opening, which prevents inward domain translocation until substrate binding takes place. Furthermore, we describe a new cation-binding site ideally suited to bind a counter-transported ion. We suggest that potassium binding at this site stabilizes the translocation-competent conformation of the unloaded transport domain in mammalian homologues.

## Introduction

Glutamate transporters, or excitatory amino acid transporters (EAATs), reside in the plasma membranes of glial cells and neurons, where they catalyze the re-uptake of the neurotransmitters glutamate and aspartate (L-asp) (**Danbolt, 2001**). EAATs terminate neurotransmission events supporting memory formation and cognition, and also prevent excitotoxicity caused by overstimulation of glutamate receptors. Dysfunction of EAATs is linked to neurological disorders, poor recovery from stroke and traumatic brain injuries (**Yi and Hazell, 2006**; **Sheldon and Robinson, 2007**; **Kim et al., 2013**). To maintain steep trans-membrane glutamate gradients, EAATs transport one substrate molecule together with three sodium ions ($Na^+$) and one proton. After their release into the cytoplasm, counter-transport of one potassium ion ($K^+$) resets the transporter for the next cycle (**Zerangue and Kavanaugh, 1996**; **Levy et al., 1998**; **Owe et al., 2006**).

Key mechanistic and structural insights into this family of transporters come from studies on an archaeal homologue from *Pyrococcus horikoshii*, $Glt_{Ph}$ (**Figure 1—figure supplement 1**), which symports L-asp together with three $Na^+$ ions (**Groeneveld and Slotboom, 2010**); however, it shows no dependence on counter-transport of $K^+$ under the conditions tested (**Ryan et al., 2009**). $Glt_{Ph}$, like EAATs, is a homo-trimer (**Gendreau et al., 2004**; **Yernool et al., 2004**). Each protomer consists of a central scaffolding trimerization domain and a peripheral transport domain containing the substrate and ion binding sites (**Boudker et al., 2007**; **Reyes et al., 2009**). When bound to $Na^+$ and L-asp ('fully bound' from here on), each transport domain moves by ~15 Å across the membrane from an outward- to an inward-facing position, in which the substrate binding site is near the extracellular solution and the cytoplasm, respectively (**Reyes et al., 2009**). Structurally symmetric helical hairpins, HP1 and HP2,

**eLife digest** Molecules of glutamate can carry messages between cells in the brain, and these signals are essential for thought and memory. Glutamate molecules can also act as signals to build new connections between brain cells and to prune away unnecessary ones. However, too much glutamate outside of the cells kills the brain tissue and can lead to devastating brain diseases.

In a healthy brain, special pumps called glutamate transporters move these molecules back into the brain cells, where they can be stored safely. However, when brain cells are damaged—by, for example, a stroke or an injury,—the glutamate stored inside spills out, killing the surrounding cells. This leads to a cascade of dying cells and leaking glutamate, which causes even more damage and slows the recovery.

Glutamate transporters ensure that there are more glutamate molecules inside cells than outside. However, it requires energy to maintain this gradient in the concentration of glutamate molecules. The transporters get this energy by moving three sodium ions into the cell with each glutamate molecule, and moving one potassium ion out of the cell. However, it is not clear how these transporters ensure that they move the glutamate molecules and the sodium ions at the same time.

Now, Verdon, Oh et al. have uncovered the 3D structure of a glutamate transporter homologue at each step of the transport process. These structures reveal that, on the outside of the cell membrane, sodium ions attach to the so-called 'transporter domain' and make it better able to bind glutamate. The transporter domain then carries the sodium ions and glutamate through the cell membrane and releases them into the cell. Verdon, Oh et al. suggest that a potassium ion then binds to the empty transport domain, stabilizing it into a more compact shape that easily makes the return trip to the outside of the cell.

Most experiments on glutamate transporters, including the work of Verdon, Oh et al., are carried out on model proteins taken from bacteria. An important challenge for the future will be to obtain structural information on human glutamate transporters, as these could be therapeutic targets for the treatment of various neurological conditions.

occlude the bound substrate from the solvent and are thought to serve as gates (*Boudker et al., 2007*; *Huang and Tajkhorshid, 2008*; *Shrivastava et al., 2008*; *Reyes et al., 2009*; *DeChancie et al., 2010*; *Focke et al., 2011*; *Zomot and Bahar, 2013*). Two $Na^+$-binding sites (Na1 and Na2), neither of which directly coordinates the substrate, were identified crystallographically using thallium ($Tl^+$) (*Boudker et al., 2007*). The location of the third $Na^+$-binding site is being debated (*Holley and Kavanaugh, 2009*; *Huang and Tajkhorshid, 2010*; *Larsson et al., 2010*; *Tao et al., 2010*; *Bastug et al., 2012*; *Teichman et al., 2012*). A highly conserved non-helical Asn310-Met311-Asp312 (NMD) motif interrupts trans-membrane segment (TM) 7 (see below). It lines the back of the substrate- and ion-binding sites and is involved in binding of the ligands (*Rosental et al., 2006*; *Tao et al., 2006*; *Rosental and Kanner, 2010*). The main chain carbonyl oxygen of Asn310 contributes to Na1 site, while the side chain of Met311 protrudes between the substrate, Na1 and Na2 binding sites (*Boudker et al., 2007*).

Symport requires that neither the substrate nor the ions alone are efficiently transported (*Crane, 1977*). Therefore to traverse the membrane, the transport domains of $Glt_{Ph}$ and EAATs must be loaded with both $Na^+$ ions and substrate. To complete the transport cycle, the transport domain of $Glt_{Ph}$ must also translocate readily when it is free of both solutes (apo), while in EAATs it requires binding of a $K^+$ ion. To establish the structural underpinnings of these processes, we determined crystal structures of the outward- and inward-facing states of $Glt_{Ph}$ in apo and ions-only bound forms (*Tables 1, 2 and 3*). We find that the apo transport domain shows identical structures when facing outward or inward. While ligand-binding sites are distorted, the domain remains compact, suggesting that it relocates across the membrane as a rigid body, similarly to when it is fully bound (*Reyes et al., 2009*). Ion binding to Na1 site, located deep in the core of the transport domain, triggers structural changes that are propagated to the extracellular gate HP2, at least in part, by the side chain of Met311 in the NMD motif. Consequently HP2, which in the apo form is collapsed into the substrate binding and Na2 sites, frees the sites, assuming conformations more similar to the conformation

**Table 1.** X-ray crystallographic data and refinement statistics for Glt$_{Ph}$-R397A and Glt$_{Ph}$-K55C-A364C$_{Hg}$ (Glt$_{Ph}$$^{in}$) structures deposited at the PDB

| | Glt$_{Ph}$$^{in}$ | | | |
|---|---|---|---|---|
| | apo | Tl$^+$-bound (apo conf.) | alkali-free | Tl$^+$-bound (bound conf.) |
| **Data collection** | | | | |
| Space group | C222$_1$ | C222$_1$ | C222$_1$ | C222$_1$ |
| Cell dimensions | | | | |
| $a$, $b$, $c$ (Å) | 109.93, 201.81, 207.14 | 106.98, 196.56, 206.50 | 106.95, 196.84, 207.48 | 110.83, 200.43, 206.40 |
| α, β, γ (°) | 90.00, 90.00, 90.00 | 90.00, 90.00, 90.00 | 90.00, 90.00, 90.0 | 90.00, 90.00, 90.00 |
| Resolution (Å) | 100.0–3.25 (3.31–3.25) | 100.0–3.75 (3.81–3.75) | 100.0–3.50 (3.56–3.50) | 100.0–4.0 (4.14–4.0) |
| $R_{sym}$ or $R_{merge}$ | 10.9 (88.6) | 14.0 (94.4) | 8.0 (88.1) | 16.3 (75.2) |
| $I/\sigma I$ | 12.3 (1.2) | 8.95 (1.1) | 13.5 (1.2) | 7.9 (1.3) |
| Completeness (%) | 98.7 (88.1) | 99.7 (99.8) | 94.4 (92.7) | 65.2 (6.5) |
| Redundancy | 5.6 (2.8) | 3.8 (3.7) | 3.3 (3.2) | 3.4 (3.5) |
| **Refinement** | | | | |
| Resolution (Å) | 15.0–3.25 | 15.0–3.75 | 15.0–3.5 | 15.0–4.0 |
| No. reflections | 34534 | 21565 | 25446 | 11105 |
| $R_{work}$/$R_{free}$ | 22.2/25.8 | 23.0/25.7 | 26.3/27.8 | 25.8/29.6 |
| No. atoms | | | | |
| Protein | 9121 | 9114 | 9088 | 8985 |
| Ligand/ion | 3 | 9 | 3 | 9 |
| $B$-factors | | | | |
| Protein | 108.5 | 141.8 | 144.2 | 137.2 |
| Ligand/ion | 135.3 | 170.8 | 214.1 | 102.3 |
| R.m.s. deviations | | | | |
| Bond lengths (Å) | 0.010 | 0.013 | 0.005 | 0.012 |
| Bond angles (°) | 1.680 | 1.861 | 1.116 | 1.407 |
| PDB code | 4P19 | 4P1A | 4P3J | 4P6H |

| | Glt$_{Ph}$-R397A | | |
|---|---|---|---|
| | Apo | Na$^+$-bound | Na$^+$/aspartate-bound |
| **Data collection** | | | |
| Space group | P2$_1$ | P3$_1$ | P3$_1$ |
| Cell dimensions | | | |
| $a$, $b$, $c$ (Å) | 112.37, 424.42, 113.99 | 110.58, 110.58, 306.92 | 116.96, 116.96, 313.52 |
| α, β, γ (°) | 90.00, 119.40, 90.00 | 90.00, 90.00, 120.00 | 90.00, 90.00, 120.00 |
| Resolution (Å) | 100.0–4.00 (4.14–4.00) | 50.0–3.39 (3.51–3.39) | 100.0–3.50 (3.63–3.50) |
| $R_{sym}$ or $R_{merge}$ | 7.8 (62.2) | 14.0 (>100) | 8.4 (>100) |
| $I/\sigma I$ | 9.3 (1.3) | 13.8 (1.4) | 10.6 (0.4) |
| Completeness (%) | 67.9 (13.0) | 87.3 (12.0) | 98.1 (96.6) |
| Redundancy | 1.8 (2.0) | 11.8 (8.6) | 4.5 (4.2) |
| **Refinement** | | | |
| Resolution (Å) | 20.0–4.0 | 12.0–3.41 | 15.0–3.50 |
| No. reflections | 52068 | 48366 | 55613 |
| $R_{work}$/$R_{free}$ | 24.9/26.6 | 28.4/29.3 | 24.3/26.8 |

*Table 1. Continued on next page*

*Table 1. Continued*

| | Glt$_{Ph}$-R397A | | |
| | Apo | Na$^+$-bound | Na$^+$/aspartate-bound |
|---|---|---|---|
| No. atoms | | | |
| Protein | 35277 | 17580 | 18192 |
| Ligand/ion | N/A | 6 | 54/12 |
| Water | N/A | 6 | 6 |
| *B*-factors | | | |
| Protein | 139.5 | 152.0 | 97.1 |
| Ligand/ion | N/A | 145.1 | 84.7/86.9 |
| Water | N/A | 102.6 | 144.6 |
| R.m.s. deviations | | | |
| Bond lengths (Å) | 0.010 | 0.010 | 0.015 |
| Bond angles (°) | 1.393 | 1.468 | 1.735 |
| PDB code | 4OYE | 4OYF | 4OYG |

**Table 2.** Completeness of datasets corrected for anisotropy

**Tl$^+$-bound Glt$_{Ph}$$^{in}$ (bound conformation)**

| Resolution range (Å) | Completeness (%) |
|---|---|
| 100.0–8.62 | 99.3 |
| 8.62–6.84 | 99.9 |
| 6.84–5.97 | 100.0 |
| 5.97–5.43 | 99.9 |
| 5.43–5.04 | 99.9 |
| 5.04–4.74 | 69.6 |
| 4.74–4.50 | 39.2 |
| 4.50–4.31 | 23.6 |
| 4.31–4.14 | 14.4 |
| 4.14–4.00 | 6.5 |

**Na$^+$-bound Glt$_{Ph}$-R397A**

| Resolution range (Å) | Completeness (%) |
|---|---|
| 50.00–7.30 | 99.6 |
| 7.30–5.79 | 100.0 |
| 5.79–5.06 | 100.0 |
| 5.06–4.60 | 100.0 |
| 4.60–4.27 | 100.0 |
| 4.27–4.02 | 100.0 |
| 4.02–3.82 | 100.0 |
| 3.82–3.65 | 98.6 |
| 3.65–3.51 | 63.0 |
| 3.51–3.39 | 12.0 |

**Apo Glt$_{Ph}$-R397A**

| Resolution range (Å) | Completeness (%) |
|---|---|
| 100.0–8.62 | 85.0 |
| 8.62–6.84 | 75.6 |
| 6.84–5.97 | 75.5 |
| 5.97–5.43 | 75.3 |
| 5.43–5.04 | 75.2 |
| 5.04–4.74 | 75.8 |
| 4.74–4.50 | 75.3 |
| 4.50–4.31 | 75.4 |
| 4.31–4.14 | 51.7 |
| 4.14–4.00 | 13.0 |

observed in the fully bound transporter. We suggest that these Na$^+$-dependent structural changes underlie the high cooperativity of Na$^+$ and substrate binding, which is thought to be one of the key coupling mechanisms (*Reyes et al., 2013*). Furthermore, in the structure of Na$^+$-bound outward-facing

**Table 3.** X-ray crystallographic data and refinement statistics for Glt$_{Ph}$-R397A and Glt$_{Ph}$-K55C-A364C$_{Hg}$ structures not deposited at the PDB

| | Glt$_{Ph}$-R397A | Glt$_{Ph}$$^{in}$ | |
| --- | --- | --- | --- |
| | Tl$^+$-bound (apo conf.) | Tl$^+$/Na$^+$ (apo conf.) | Tl$^+$/k$^+$ (apo conf.) |
| **Data collection** | | | |
| Space group | $P2_1$ | $C222_1$ | $C222_1$ |
| Cell dimensions | | | |
| a, b, c (Å) | 115.18, 428.53, 116.61 | 108.11, 198.86, 206.34 | 106.59, 198.48, 205.82 |
| α, β, γ (°) | 90.00, 119.49, 90.00 | 90.00, 90.00, 90.00 | 90.00, 90.00, 90.00 |
| Resolution (Å) | 30.0–5.0 (5.18–5.00) | 100.0–4.0 (4.07–4.00) | 100.0–4.15 (4.22–4.15) |
| $R_{sym}$ or $R_{merge}$ | 10.9 (>100) | 15.0 (92.2) | 13.9 (94.1) |
| $I/\sigma I$ | 13.8 (1.9) | 8.9 (1.5) | 9.2 (1.5) |
| Completeness (%) | 86.4 (75.1) | 99.9 (100) | 94.5 (90.2) |
| Redundancy | 5.5 (5.8) | 3.9 (3.9) | 4.0 (3.9) |
| **Refinement** | | | |
| Resolution (Å) | 20.0–5.0 | 15.0–4.0 | 15.0–4.15 |
| No. reflections | 34747 | 18184 | 15419 |
| $R_{work}$/$R_{free}$ | 22.0/26.5 | 28.2/31.7 | 28.3/31.2 |
| No. atoms | | | |
| Protein | 35107 | 9135 | 9135 |
| Ligand/ion | N/A | N/A | N/A |
| Water | N/A | N/A | N/A |
| B-factors | | | |
| Protein | 223.00 | 183.6 | 194.4 |
| Ligand/ion | N/A | N/A | N/A |
| Water | N/A | N/A | N/A |
| R.m.s. deviations | | | |
| Bond lengths (Å) | 0.008 | 0.006 | 0.008 |
| Bond angles (°) | 1.186 | 1.266 | 1.440 |

Glt$_{Ph}$ we observe opening of HP2 tip, which may facilitate L-asp access to its binding site and prevent the inward movement of the Na$^+$-only bound transport domain, as previously suggested (*Focke et al., 2011*). Remarkably, soaks of apo Glt$_{Ph}$ crystals in Tl$^+$ reveal new cation-binding sites within the apo-like protein architecture. One such site overlaps with the substrate-binding site. Because binding of a cation to this site would compete with binding of Na$^+$ and the transported substrate, it is well suited to serve as a binding site for a counter-transported ion. We propose that the closed translocation-competent conformation of the transport domain free of Na$^+$ and substrate is intrinsically stable in Glt$_{Ph}$ but not in EAATs, in which K$^+$ binding at the newly identified site is required, coupling transport cycle completion to K$^+$ counter-transport.

## Results

### Remodeling of the apo transport domain

To determine the structure of apo Glt$_{Ph}$, we used R397A mutant that shows a drastically decreased affinity for substrate (*Figure 1A*). When fully bound, Glt$_{Ph}$-R397A crystallizes in the outward-facing state, like wild type Glt$_{Ph}$, except that L-asp coordination is slightly altered because the mutant is missing the key coordinating side chain of Arg397 (*Figure 1B*, *Figure 1—figure supplement 2*; *Bendahan et al., 2000*; *Boudker et al., 2007*). These results suggest that R397A is suitable to capture the apo and ions-only bound outward-facing states for their structural characterization.

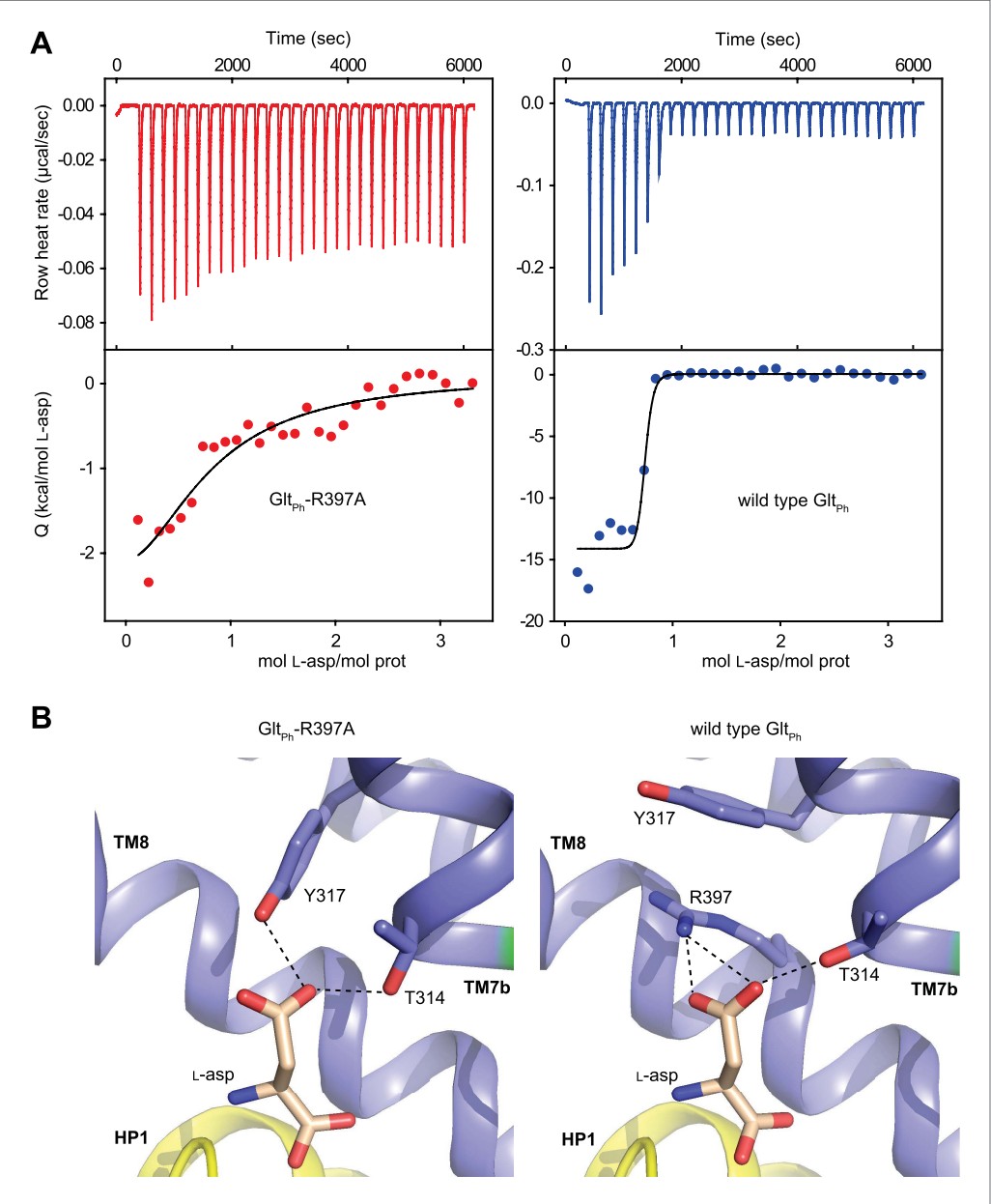

**Figure 1**. Substrate binding to Glt$_{Ph}$-R397A. (**A**) Raw binding heat rates measured by isothermal titration calorimetry (top) and binding isotherms (bottom) obtained for Glt$_{Ph}$-R397A (left) and wild type Glt$_{Ph}$ (right) at 25°C in the presence of 100 mM NaCl. The solid lines through the data are fits to the independent binding sites model with the following parameters for Glt$_{Ph}$-R397A and wild type Glt$_{Ph}$, respectively: enthalpy change ($\Delta H$) of −3.2 and −14.3 kcal/mol; the apparent number of binding sites ($n$) of 0.8 and 0.7 per monomer; dissociation constant ($K_d$) of 6.6 µM and 27 nM. Note that L-asp binding to the wild type transporter is too tight at 100 mM NaCl to be accurately measured in this experiment. The binding $K_d$ has been estimated to be ~1 nM (**Boudker et al., 2007**). (**B**) L-asp binding site in Glt$_{Ph}$-R397A (left) and wild type Glt$_{Ph}$ (right). L-asp and residues coordinating the side chain carboxylate are shown as sticks with carbon atoms colored light brown and blue, respectively. Potential hydrogen bonds (distances less than 3.5 Å) between the L-asp side chain carboxylate and transporter residues are shown as dashed lines. Note that Y317, which forms cation-π interactions with guanidium group of R397 in wild type Glt$_{Ph}$, interacts directly with L-asp in Glt$_{Ph}$-R397A.

The following figure supplements are available for figure 1:

**Figure supplement 1**. Alternating access mechanism in Glt$_{Ph}$.

**Figure supplement 2**. Structure of Glt$_{Ph}$-R397A bound to Na$^+$ and L-asp.

However, removal of Arg397 may affect local electrostatics, potentially altering ion binding; thus these studies should be interpreted with caution. Apo Glt$_{Ph}$-R397A also crystallized in an outward-facing conformation that is similar to the structure reported for a close Glt$_{Ph}$ homologue (*Jensen et al., 2013*). To obtain an apo inward-facing state, we used Glt$_{Ph}$-K55C-A364C mutant trapped in the inward-facing state upon cross-linking with mercury (*Reyes et al., 2009*) (Glt$_{Ph}$$^{in}$, *Figure 1—figure supplement 1*). The positions and orientations of the transport domains relative to the trimerization domains remain essentially unchanged in the apo and fully bound forms of Glt$_{Ph}$-R397A and Glt$_{Ph}$$^{in}$ (*Figure 2*). In contrast, the conformations of the transport domains themselves differ significantly. Most remarkably, the apo conformations of the transport domain are nearly identical in the outward- and inward-facing states (*Figure 3A*, *Figure 3—figure supplement 1*, *Figure 3—figure supplement 2A*) and are therefore independent of the transport domain orientations and crystal packing environments.

The conformational differences between fully bound and apo forms of the transport domain include a concerted movement of HP2 and TM8a, which form the extracellular surface of the domain, and local rearrangements at the ligand binding sites, involving HP2, the NMD motif and TM3 (*Figure 3B–E*, *Figure 3—figure supplement 2B*, *Figure 4*). In HP2, the last helical turn of HP2a unwinds, and HP2a together with the loop region at HP2 tip collapse into the substrate and Na2 binding sites. Within the NMD motif, the side chain of Asn310 rotates away from TM3 and partially fills the empty Na1 site, while the side chain of Met311 undergoes an opposite movement, flipping away from the binding sites (*Figure 4*). Finally, TM3 bends away from the NMD motif, particularly around Thr92 and Ser93 (*Figure 3B,C*). Notably, these residues together with the side chain of Asn310 form one of the proposed third Na$^+$-binding sites (*Huang and Tajkhorshid, 2010*; *Bastug et al., 2012*). Thus, all known ligand-binding sites are distorted in the apo forms (*Figure 4*).

The overall structures of the apo transport domain remain as closed and compacted as in the fully bound forms (*Figure 4—figure supplement 1*). Therefore, we propose that the unloaded transport domains traverse the membrane as rigid bodies as deduced previously for the fully loaded transport domains (*Reyes et al., 2009*).

## Insight into the coupling mechanism

In Glt$_{Ph}$, cooperative binding of Na$^+$ ions and L-asp is central to tightly coupled transport of the solutes (*Reyes et al., 2013*). Our structures of the apo and fully bound Glt$_{Ph}$ suggest that binding of L-asp and

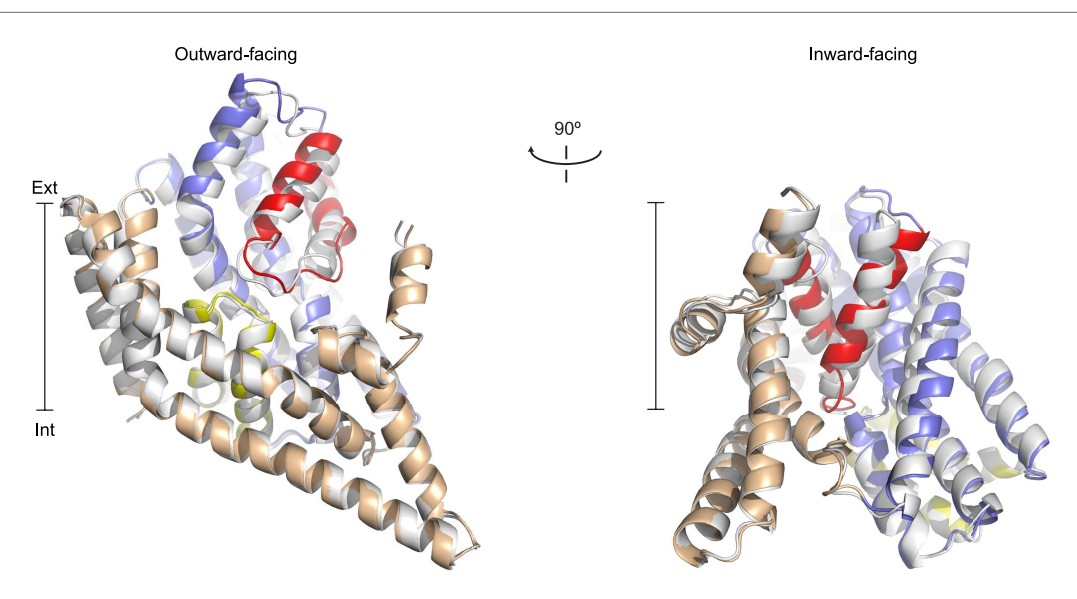

**Figure 2**. Apo protomer structures. (**A**) Glt$_{Ph}$ protomers in the outward-facing state (left) and a Glt$_{Ph}$$^{in}$ protomer (right) viewed from within the plane of the membrane. Shown are superimpositions between apo (colors) and fully bound protomers (grey).

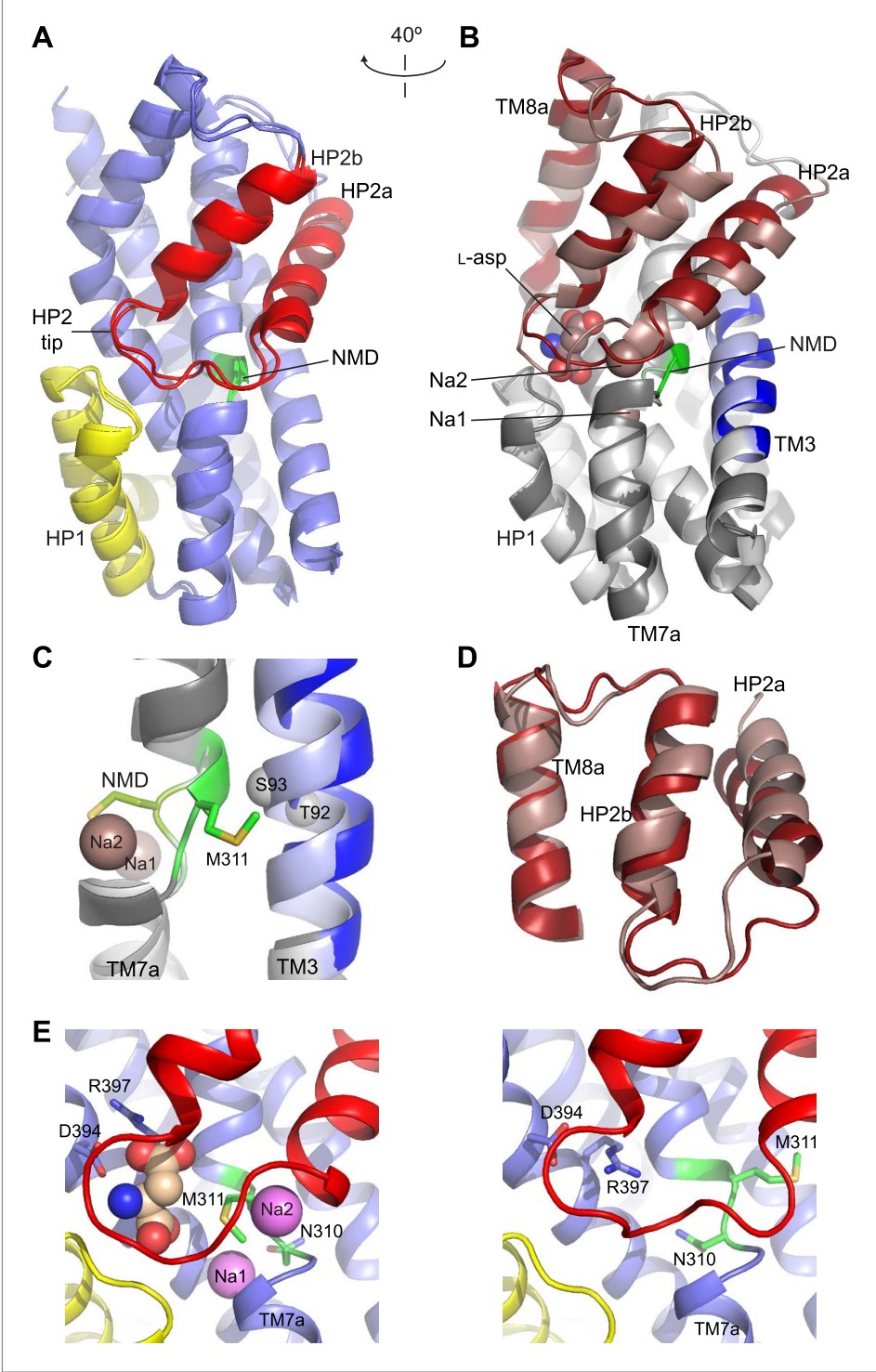

**Figure 3**. Structures of the apo transport domain. (**A**) Superimposition of the nearly identical apo transport domains in the outward- and inward-facing states. HP1, HP2, and NMD motif are colored yellow, red, and green, respectively. The remainder of the domain is blue. (**B**) Superimposition of the fully bound (light colors, PDB accession number 2NWX) and apo Glt$_{Ph}$$^{in}$ (dark colors) transport domains. (**C**) The NMD motif and adjacent TM3. Met311 is shown as sticks, and the light blue spheres indicate the Cα positions for T92 and S93. (**D**) The HP2-TM8a structural modules in the fully bound (pink) and apo (red) transport domains superimposed on TM8a and HP2b to emphasize the re-orientation of the HP2a. (**E**) The Na$^+$ and L-asp binding sites in the fully bound (left) and apo forms (right).

*Figure 3. Continued on next page*

*Figure 3. Continued*

The following figure supplements are available for figure 3:

**Figure supplement 1**. Apo protomer structures.

**Figure supplement 2**. Structural comparison of the transport domain in various states.

---

Na$^+$ at the Na2 site is coupled because the same structural element, the tip of HP2, contributes to both sites and is restructured upon binding. Thus, structural changes in HP2 upon binding of either L-asp or Na$^+$ ion should greatly favor binding of the other.

Met311 in the NMD motif is the only residue that is shared between the Na1 site and the substrate and Na2 sites and also undergoes a conformational change upon ligand binding. To examine whether the structural changes in HP2 upon binding of L-asp and Na$^+$ at the Na2 site could occur independently from those in the NMD motif upon Na$^+$ binding at the Na1 site, we modeled transport domains with HP2 in the bound conformation and the NMD motif in the apo conformation, or *vice versa* (*Figure 5A*). In both models, the side chain of Met311 clashes with residues in HP2, suggesting that the conformational changes in HP2 and the NMD motif must be concerted.

We then mutated bulky Met311 to either another bulky residue, leucine, or to a smaller residue, alanine, which is not expected to experience similar clashes. For these mutants, generated in the context of unconstrained wild type Glt$_{Ph}$ and inward cross-linked Glt$_{Ph}$$^{in}$, we measured the dependence of L-asp dissociation constant on Na$^+$ concentration (*Figure 5B*). While this dependence is very steep for the wild type Glt$_{Ph}$ constructs (*Reyes et al., 2013*) and nearly as steep for the M311L mutants, it is substantially shallower for the M311A mutants. The most parsimonious interpretation of these results is that M311A mutation reduces binding cooperativity between the substrate and Na$^+$ ions. However, it is also possible, though we think unlikely, that the mutation abrogates ion binding at one or more Na$^+$-binding sites in the tested concentration range (1–100 mM). Mutating the equivalent methionine to smaller residues in EAAT3 also resulted in less steep dependence of the ionic currents on Na$^+$ concentration (*Rosental and Kanner, 2010*). Based on these results, we hypothesize that Met311 is key to the allosteric coupling between the Na1, L-asp and Na2 sites. Consistently, bulky methionine or leucine residues are found at this position in ~85% of glutamate transporter homologues. However, it should be noted that methionine is conserved in the Na$^+$-coupled Glt$_{Ph}$ and EAATs, while a characterized proton-coupled homologue has leucine at this position (*Gaillard et al., 1996*). Hence, it is possible that the methionine thioether, which is proximal to both Na1 and Na2 sites, plays a direct role in Na$^+$ binding.

Our hypothesis further predicts that binding of an ion at Na1 site should prime the transporter to accept its substrate. Therefore, we crystallized Glt$_{Ph}$-R397A in the presence of 400 mM Na$^+$, but in the absence of L-asp. We also soaked crystals of apo Glt$_{Ph}$$^{in}$ in Tl$^+$, an ion with strong anomalous signal that seems to mimic some aspects of Na$^+$ in Glt$_{Ph}$ and EAATs (*Boudker et al., 2007*; *Tao et al., 2008*). The obtained outward- and inward-facing structures pictured the transport domains in conformations overall similar to those observed in the fully bound transporter: straightened TM3, Met311 pointing toward the binding sites, extended helix in HP2a and HP2 tip raised out of the substrate binding site (*Figure 6A–D*). Indeed, the structure of Tl$^+$-bound Glt$_{Ph}$$^{in}$ is indistinguishable from the fully bound Glt$_{Ph}$$^{in}$ and both Na1 and Na2 sites are occupied by Tl$^+$ ions (*Figure 6A*). The structure of Na$^+$-bound Glt$_{Ph}$-R397A differs significantly from the fully bound Glt$_{Ph}$-R397A only at the tip of HP2 (*Figure 3—figure supplement 2*, also see below). The coordinating residues at the Na1 site are correctly positioned and the site is likely occupied by a Na$^+$ ion. The Na2 site still shows a distorted geometry: the last helical turn of HP2a points away from the site due to the altered conformation of the tip of HP2 (*Figure 6C*). Collectively, our results demonstrate that binding of the coupled ions, notably at the Na1 site, is sufficient to trigger isomerization of the transport domain from the apo conformation to the bound-like conformation. The energetic penalty associated with this isomerization likely explains why Na$^+$ ions alone bind weakly to the transporter (*Reyes et al., 2013*). This experimental observation contrasts with highly favorable calculated binding energies (approximately −10 kcal/mol for Na1) that were obtained using fully bound protein conformation and where the reference ion-free state is the same as the bound state (*Larsson et al., 2010*; *Bastug et al., 2012*; *Heinzelmann et al., 2013*).

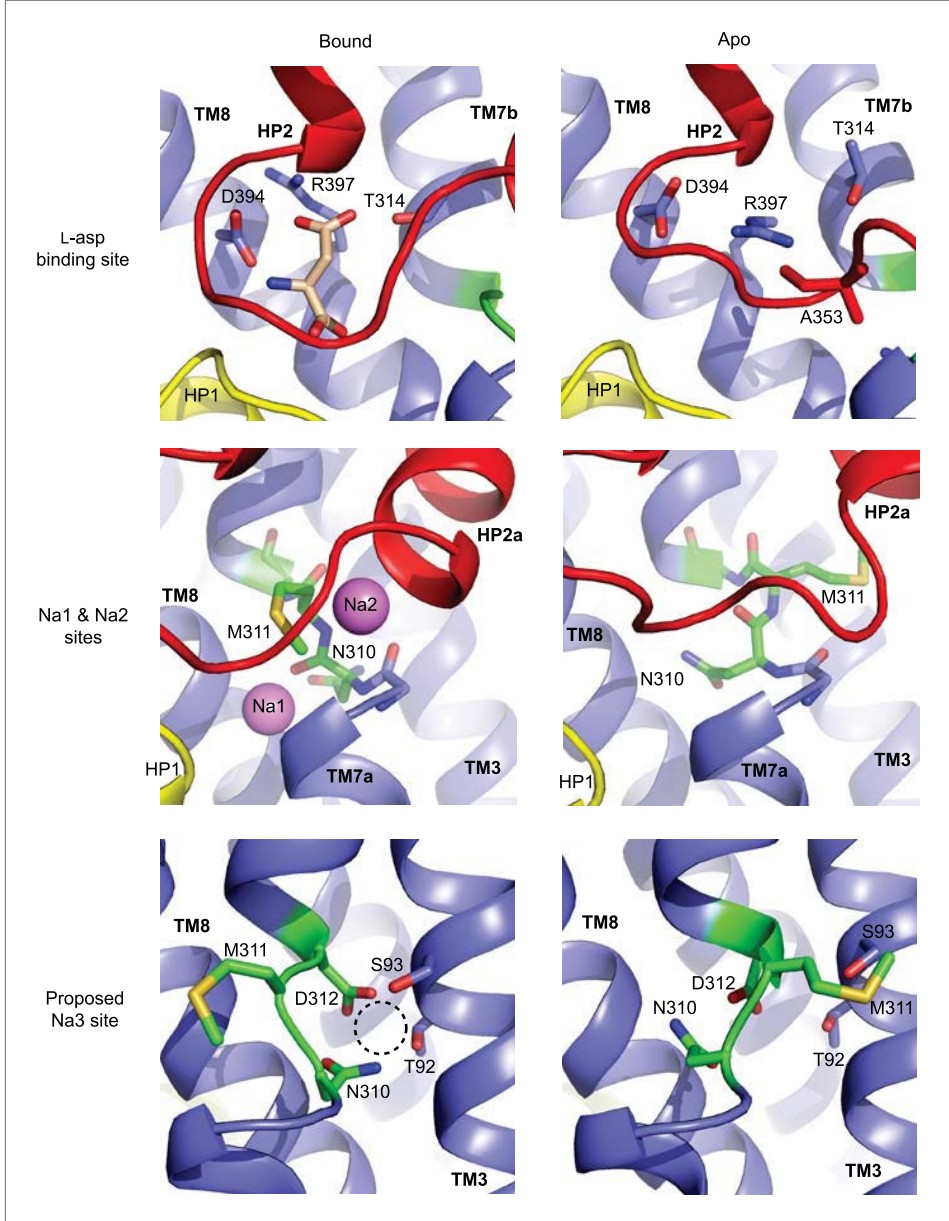

**Figure 4**. Remodeling of L-asp and Na⁺ binding sites in the apo conformations. Close-up views of the fully bound (left) and apo (right) transport domains at L-asp binding site (top), Na1 and Na2 sites (middle), and one of the proposed locations for the third Na⁺ binding site (**Huang and Tajkhorshid, 2010**; **Bastug et al., 2012**) (dashed circle).

The following figure supplements are available for figure 4:

**Figure supplement 1**. Transport domain remains compact.

## Na⁺-mediated gating in the outward-facing state

The structure of the Na⁺-only bound Glt$_{Ph}$-R397A shows HP2 in a conformation overall similar to that observed in the fully bound transporter, but with an opened tip (**Figure 6B–D,F**, **Figure 6—figure supplement 1**). This opening is smaller than the opening observed previously in the structure of Glt$_{Ph}$ in complex with the blocker L-threo-β-benzyloxyaspartate (**Figure 6—figure supplement 2**; **Boudker et al., 2007**), and it is hinged at two well-conserved glycine residues at positions 351 and 357 (**Figure 6C**). Interestingly, among the nine amino acids forming the tip in Glt$_{Ph}$ (residues 351 to 359), five are glycines in the consensus sequence generated for the glutamate transporter family, although not all are present

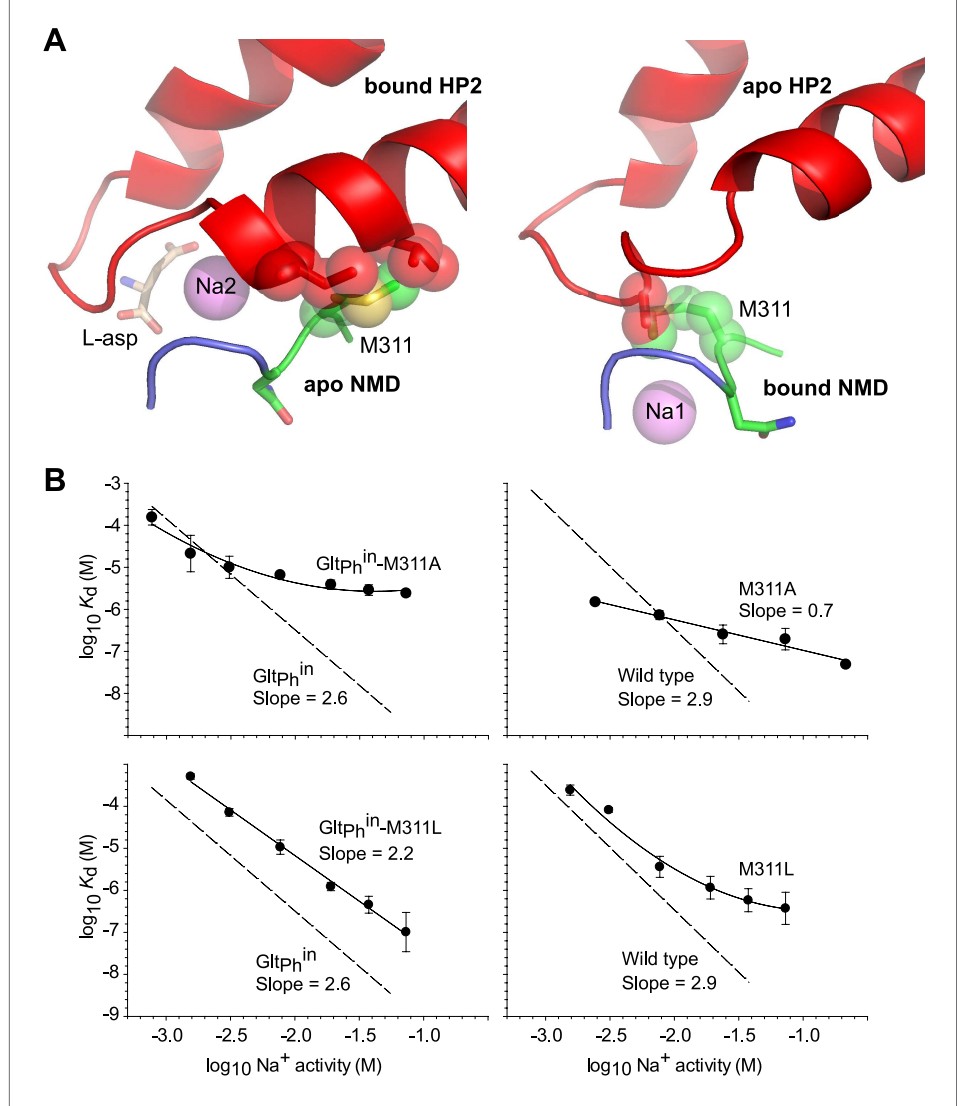

**Figure 5**. Met311 is key to the allosteric coupling. (**A**) Structural models combining HP2 bound to L-asp and $Na^+$ at Na2 site with apo conformation of the NMD motif (left), and apo conformation of HP2 with the NMD motif bound to $Na^+$ at Na1 site (right). Met311 and clashing residues in HP2 are shown as sticks and transparent spheres. (**B**) The dependence of L-asp dissociation constant, $K_d$, on $Na^+$ activity plotted on a log–log scale for mutants within the context of $Glt_{Ph}{}^{in}$ (left) and unconstrained $Glt_{Ph}$ (right). The data were fitted to straight lines with slopes shown on the graph or to arbitrary lines for clarity. Dashed lines and corresponding slopes correspond to published dependences for $Glt_{Ph}{}^{in}$ and $Glt_{Ph}$ (**Reyes et al., 2013**).

in each homologue (**Figure 6E**, **Figure 6—figure supplement 3**). We suggest that the glycines support the structural flexibility of the HP2 tip in all members of the family, but that the structural specifics of the tip opening may vary among homologues.

To test whether the trans-membrane movement of the transport domain is possible when the tip of HP2 is opened, we modeled the open tip conformation in the context of the previously reported early transition intermediate structure (**Figure 7**; **Verdon and Boudker, 2012**). In this structure, the transport domain tilts towards the trimerization domain but does not yet undergo a significant translation toward the cytoplasm. We find that such intermediate state with the opened tip of HP2 can be achieved without major steric clashes, while further progression of the transport domain to the inward-facing position could be impeded because the tip is likely to clash with TM5 in the trimerization domain (**Figure 7B**). Also in the inward-facing state HP2 is packed against the trimerization domain and cannot

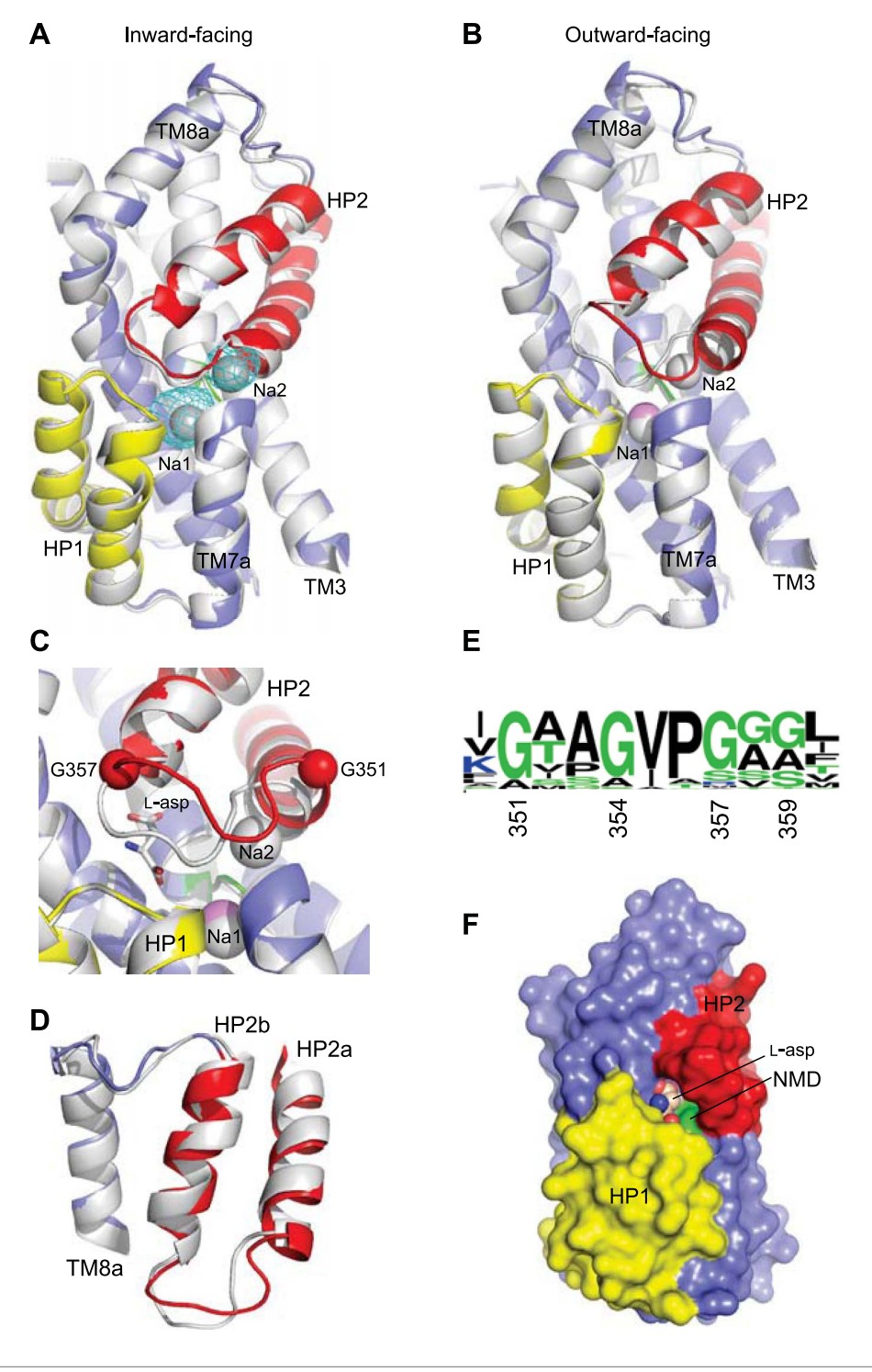

**Figure 6**. Structures of ions-only bound transport domain. (**A**) Superimposition of the fully bound transport domains (grey) and Tl$^+$-bound Glt$_{Ph}$$^{in}$ transport domain in the bound-like conformation (colors), with the averaged anomalous difference Fourier map contoured at 8σ (cyan mesh). (**B**) Superimposition of the fully bound (grey) and Na$^+$-only bound Glt$_{Ph}$-R397A (colors) transport domains. (**C**) Na$^+$ and L-asp binding sites with fully-bound structure shown in white and Na$^+$-bound structure in colors. Hinge glycine residues are shown as spheres. The modeled Na$^+$ ion in Na1 site is pink. (**D**) Superimposition of the HP2-TM8 in the fully bound transport domain (grey) and in Glt$_{Ph}$-R397A bound to Na$^+$ only (colors), showing similar conformations of HP2a. (**E**) WebLogo representation of the
*Figure 6. Continued on next page*

*Figure 6. Continued*

consensus sequence and relative abundance of residues in HP2 tip. (**F**) Surface representation of the transport domain of Glt$_{Ph}$-R397A bound to Na$^+$ only showing access to the substrate-binding site. L-asp was placed into the binding site for reference.

The following figure supplements are available for figure 6:

**Figure supplement 1**. Na$^+$ only bound Glt$_{Ph}$-R397A.

**Figure supplement 2**. Superimposition of the transport domains bound to Na$^+$ and L-asp (light grey), Na$^+$ and L-TBOA (dark grey) and Na$^+$ only (colors).

**Figure supplement 3**. Sequence alignment for the HP2 tip region of Glt$_{Ph}$ and human EAAT sub-types 1–5.

open in a manner observed in the outward-facing state. Consistently, HP2 is closed in Glt$_{Ph}$$^{in}$ bound to Tl$^+$ (*Figure 6A*).

Therefore, opening of the HP2 tip upon Na$^+$ binding in the outward-facing state may serve as a structural mechanism preventing uncoupled uptake of Na$^+$ ions. We suggest that the structural changes in the NMD motif and HP2 that are triggered upon Na$^+$ binding at the Na1 site may lead to the loss of direct interactions between the tip of HP2 and the rest of the transport domain, resulting in tip opening. Subsequent binding of L-asp and Na$^+$ at the Na2 site is then required to provide compensatory interactions, allowing HP2 tip to close. Similar conformational behavior has been observed for transporters with the LeuT fold: when bound to Na$^+$ ions only, substrate binding sites are open to the extracellular solution, and substrate binding is required for occlusion (*Weyand et al., 2008*; *Krishnamurthy and Gouaux, 2012*).

We do not see a transition into an open conformation in the inward-facing Glt$_{Ph}$$^{in}$ bound to Tl$^+$ ions (*Figure 6A*). This may be because Tl$^+$ ions do not faithfully mimic Na$^+$ ions and fail to induce an open state or it may be because Na$^+$ bound inward-facing state is, indeed, closed. This latter possibility does not contradict the requirements of symport because the measured dissociation constant for Na$^+$ ions in the inward-facing state (250 mM) (*Reyes et al., 2013*), is far above Na$^+$ concentration in the cytoplasm (10 mM) and therefore, Na$^+$-bound inward-facing state is not expected to be significantly populated.

We and others have proposed that transition intermediates mediate fluxes of polar solutes, including anions, because potentially hydrated cavities form in such intermediates at the interface between the trimerization and transport domains (*Stolzenberg et al., 2012*; *Verdon and Boudker, 2012*; *Li et al., 2013*). Interestingly, because the tip of HP2 forms part of this interface in the fully bound intermediate state of the transporter, opening of the tip in the Na$^+$-only bound form may increase solvent accessibility to the interface (*Figure 7A*).

## New cation-binding sites

While soaking apo Glt$_{Ph}$$^{in}$ crystals in Tl$^+$ solutions, we observed that only in approximately one third of crystals Tl$^+$ ions bound to the Na1 and Na2 sites, inducing transition from apo- to bound-like conformation as described above. In the majority of the crystals, we observed no conformational changes of the transport domain and Tl$^+$ ions incorporated at two previously uncharacterized sites (*Figure 8*), within the small cavities that remain under the collapsed HP2 (*Figure 8—figure supplement 1*). One site, termed Na2', involves residues of HP2 and TM7a that form the Na2 site, but in a different ion coordinating geometry due to the conformational difference in HP2 (*Figure 8B*). The second site, termed Ct, overlaps with the L-asp binding site and is formed by the side chains of highly conserved Asp394 and Thr398 in TM8 and main chain carbonyl oxygen atoms of HP1 and HP2 (*Figure 8B,C*). Tl$^+$ soaks of the outward-facing apo Glt$_{Ph}$-R397A also showed no conformational changes of the transport domain, with Tl$^+$ binding at the Ct site, but not at the Na2' site (*Figure 8A*). The ion selectivity of the Ct site remains ambiguous, because neither 300 mM K$^+$ nor 10 mM Na$^+$ efficiently inhibited incorporation of Tl$^+$ (150 mM) at this site in Glt$_{Ph}$$^{in}$ (*Figure 8—figure supplement 2*). Crystals deteriorated at higher Na$^+$ concentrations. In contrast, the Na2' site seems to show a preference for Na$^+$, which even at low concentration (10 mM) interfered significantly with Tl$^+$ binding.

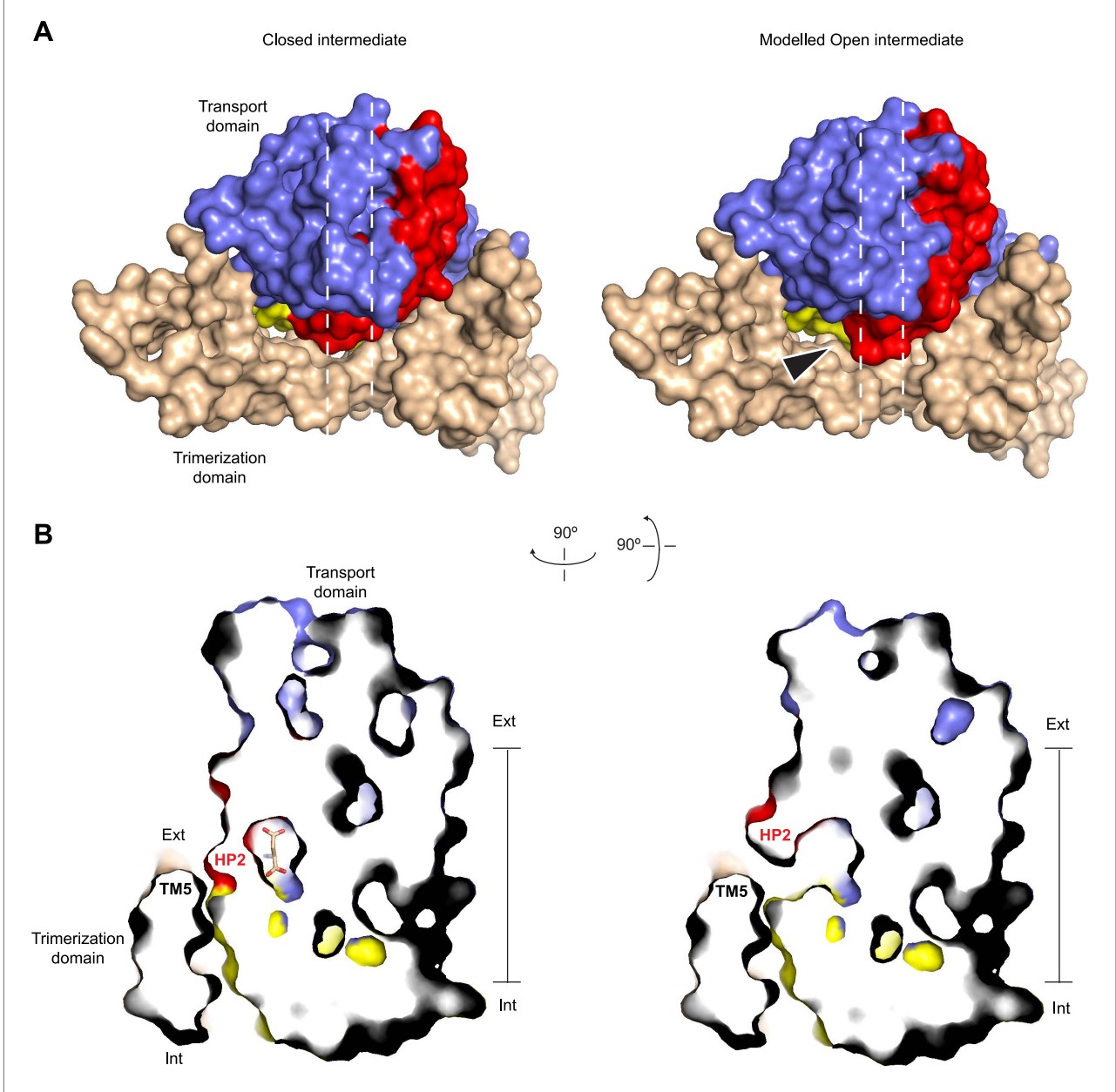

**Figure 7**. Modeled Na$^+$-bound early transition intermediate between the outward- and inward-facing states. (**A**) Surface representations of the protomer in the fully bound intermediate state (PDB code 3V8G) (left), and the modeled Na$^+$-bound intermediate with an open HP2 tip (right) viewed from the extracellular space (top). The model reveals no clashes, suggesting that the observed opening of HP2 is structurally compatible with the intermediate orientation of the transport domain. The arrows indicate the point of access to the domain interface with potentially increased solvent accessibility. (**B**) Side views of thin cross-sections of the closed fully bound (left) and open Na$^+$-only bound (right) intermediate state. The protomers are sliced normal to the membrane plane, as indicated by the dashed lines in **A**.

The functional relevance of these sites is speculative at present. However, it is remarkable that the Ct site is positioned exactly at the same place as the amino group of the bound L-asp and share several coordinating moieties. Therefore, binding of a cation at the Ct site and binding of the substrate are mutually exclusive. Because the Ct site is observed only in the apo-like conformation, cation binding at this site would also inhibit the transition into the bound-like conformation upon Na$^+$ binding at the Na1 site. Finally, the Ct site is observed in both the inward- and outward-facing states, suggesting that the apo-like transport domain could carry the ion across the membrane. These are the exact properties

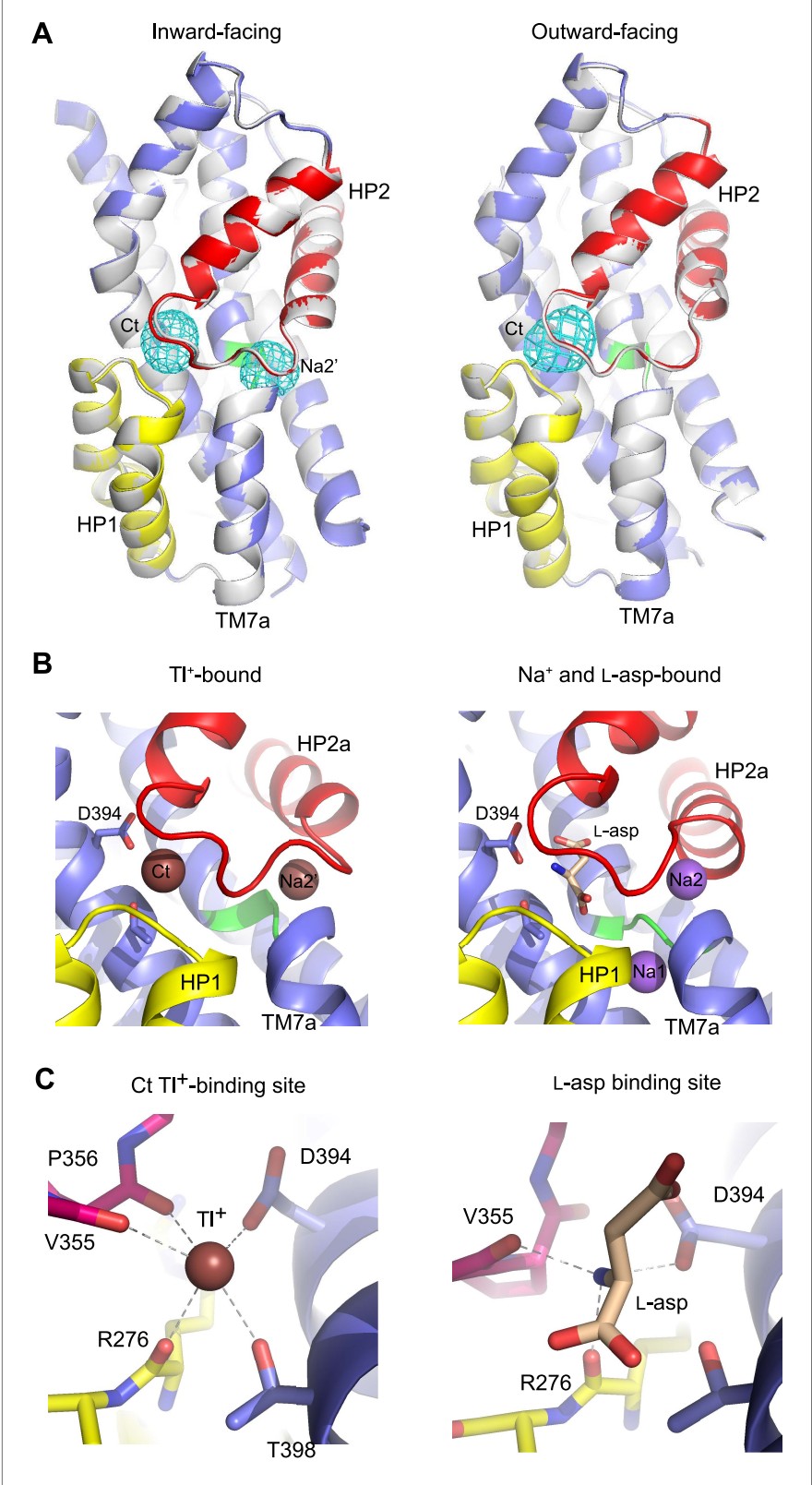

**Figure 8**. New cation binding sites. (**A**) Superimpositions of Glt$_{Ph}$$^{in}$ (left) and outward-facing Glt$_{Ph}$-R397A (right) transport domains in the apo form (grey) with Tl$^+$-bound apo-like conformations (colors). Averaged anomalous difference Fourier maps are contoured at 8σ (cyan mesh). (**B**) Modeled Tl$^+$ ions bound to the Ct and Na2' sites (left),
*Figure 8. Continued on next page*

*Figure 8. Continued*

and L-asp aspartate and Na$^+$ ions bound to the Na1 and Na2 sites in the fully bound transport domain (right). (**C**) Close-up view of Tl$^+$ in the Ct site of apo-like Glt$_{Ph}$$^{in}$ and L-asp in the fully bound transporter.

The following figure supplements are available for figure 8:

**Figure supplement 1**. Transport domain internal cavities.

**Figure supplement 2**. Specificity of the new cation binding sites in apo-like Glt$_{Ph}$$^{in}$.

expected for the K$^+$-binding site in EAATs. Moreover, it has been previously proposed that K$^+$ binds to EAATs at a similar position (*Holley and Kavanaugh, 2009*). Most remarkably, in an insect K$^+$-independent dicarboxylate transporter, an asparagine to aspartate mutation at the position equivalent to Asp394 in Glt$_{Ph}$ changes the transporter substrate specificity to amino acid glutamate, and also leads to dependence on K$^+$ counter-transport (*Wang et al., 2013*). Therefore this aspartate plays a key role in both binding the amino group of substrate and coupling to K$^+$ counter-transport. Consistently, Asp394 in Glt$_{Ph}$ coordinates both the amino group of the bound substrate and Tl$^+$ in the Ct site. Notably, while Tl$^+$ mimics, to some extent, Na$^+$ ions in Glt$_{Ph}$ and EAATs, it is a better mimic of K$^+$ ions in EAATs (*Boudker et al., 2007*; *Tao et al., 2008*).

## Movement of HP1 in the inward-facing state

To examine whether a complete removal of Na$^+$ and K$^+$ ions had an effect on the structure of Glt$_{Ph}$$^{in}$, we soaked apo Glt$_{Ph}$$^{in}$ crystals (typically grown in the presence of K$^+$) in alkali-free buffer. Interestingly, we observed a small, but reproducible structural change in several crystals examined: HP1 and TM7a that form the transport domain cytoplasmic surface moved slightly towards TM8, with the tip of HP1 detaching from that of HP2 (*Figure 9*, *Figure 9—figure supplement 1*). This movement is observed clearly in one protomer (chain B in 4P3J), in which these helices are not involved in crystal packing contacts. It is reminiscent of the isomerization of the structurally symmetric HP2 and TM8a on the extracellular side of the domain observed upon the transition from bound to apo forms (*Figure 9B*). It was suggested previously that HP1 participates in intracellular gating in Glt$_{Ph}$ (*Reyes et al., 2009*; *DeChancie et al., 2010*). Indeed, the observed movement of HP1 generates a small opening, leading to the substrate and Ct sites (*Figure 9—figure supplement 2*), and it is reminiscent of the movement observed in molecular dynamics simulations (*DeChancie et al., 2010*; *Zomot and Bahar, 2013*). However, this conformational difference is too small to be interpreted unambiguously.

## Discussion

Our apo and ions-only bound structures reveal a remarkable structural plasticity of Glt$_{Ph}$ transport domain that is likely a conserved feature in the glutamate transporter family. In addition to the large trans-membrane rigid-body movements of the transport domain between outward- and inward-facing orientations, local conformational changes within the domain accompany binding and release of the transported substrate and ions (*Figure 10*). These local changes provide a structural explanation of how Na$^+$ gradients are harnessed to drive concentrative substrate uptake, supporting two previously proposed coupling mechanisms (*Focke et al., 2011*; *Reyes et al., 2013*): allosterically coupled binding of the substrate and symported Na$^+$ ions, and opening of HP2 upon Na$^+$ binding, which impedes the inward trans-membrane movement of the Na$^+$-only bound transport domain.

The apo transport domains in the outward- and inward-facing states are essentially identical and as compact as when they are fully bound, consistent with previous spectroscopic experiments (*Focke et al., 2011*). Therefore, the apo transport domain is likely able to transition readily between the cytoplasmic and extracellular orientations. Consistently, previous spectroscopic studies showed that the transport domains continuously sample the outward- and inward-facing positions with nearly equal probabilities either when bound to Na$^+$ and L-asp or when free of the solutes (*Akyuz et al., 2013*; *Erkens et al., 2013*; *Georgieva et al., 2013*; *Hanelt et al., 2013*). Moreover, these transitions are more frequent in the apo transporter, consistent with a lack of large energetic barriers (*Akyuz et al., 2013*). In Glt$_{Ph}$, the compact translocation-competent apo conformation of the transport domain is stabilized by interactions between the collapsed HP2, and HP1, TM7, and TM8. In EAATs, by contrast, we speculate that these interactions are insufficient and that K$^+$ binding to the Ct site is required to

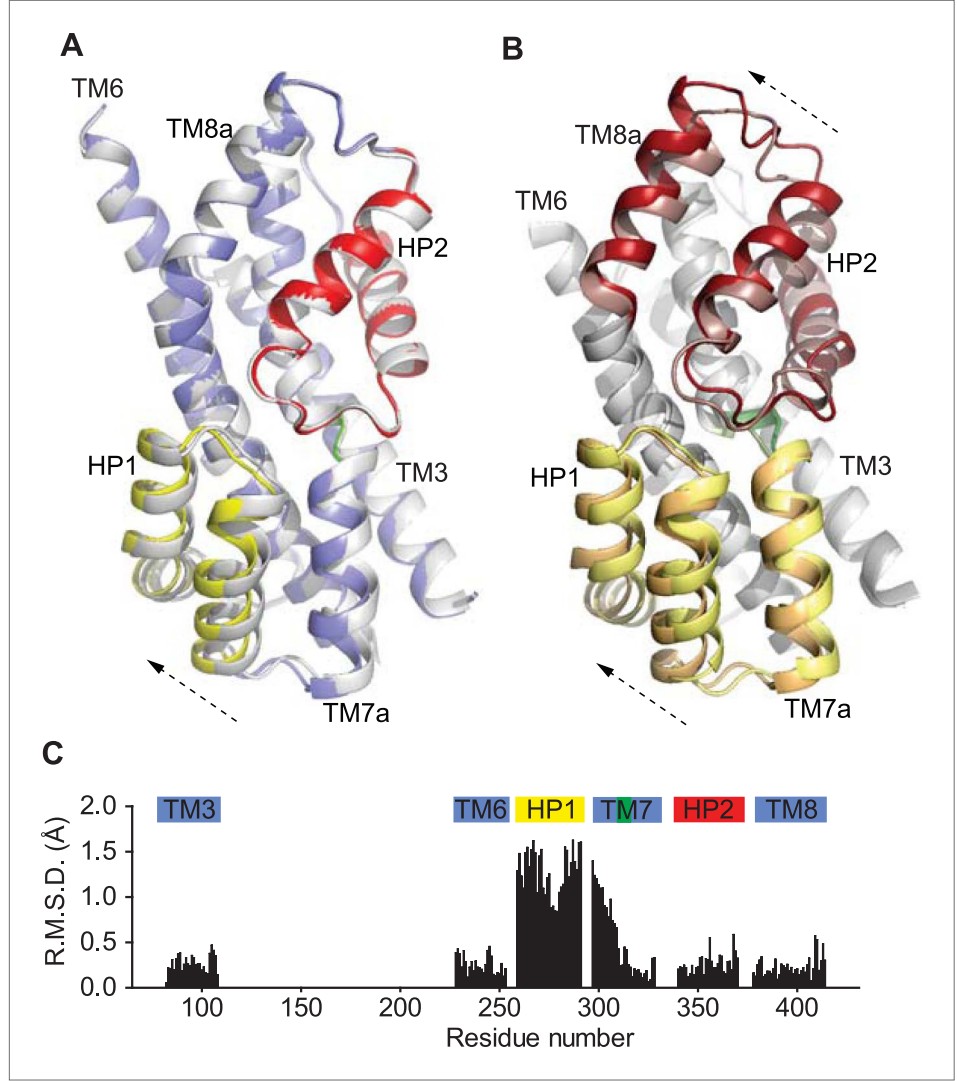

**Figure 9**. Movements of the HP1-TM7a structural module. (**A**) Superimposition of Glt$_{Ph}$$^{in}$ transport domains when bound to Tl$^+$ in the apo-like conformation (grey) and when prepared in an alkali-free solution (colors). (**B**) The transport domains of the fully bound Glt$_{Ph}$ (light colors) and alkali-free Glt$_{Ph}$$^{in}$ (dark colors) superimposed on TM6. Arrows indicate movements of the structurally symmetric HP1-TM7a and HP2-TM8a modules. (**C**) Per residue main chain R. M. S. D. values calculated for the structures of the inward-facing transport domains bound to Tl$^+$ in apo-like conformation and alkali-free shown in **A** and superimposed on HP2. The bars above the plot represent secondary structure elements colored as in **A**.

The following figure supplements are available for figure 9:

**Figure supplement 1**. Alkali-free inward-facing Glt$_{Ph}$$^{in}$.

**Figure supplement 2**. Surface representation of the alkali-free inward-facing Glt$_{Ph}$$^{in}$ transport domain in this protomer after refinement.

stabilize the translocation-competent closed conformation that can return to the outside, ensuring coupling between substrate uptake and counter-transport of K$^+$ ion. Local structural differences in EAATs in the vicinity of the Ct site may underlie the higher affinity and specificity of this site for K$^+$ ion.

In conclusion, we have shown structurally that ion binding and unbinding events in Glt$_{Ph}$ and, by analogy, in EAATs control the conformational state of the transporter, determining its competence to bind substrate and undergo transitions between the outward- and inward-facing states. Studies establishing the location

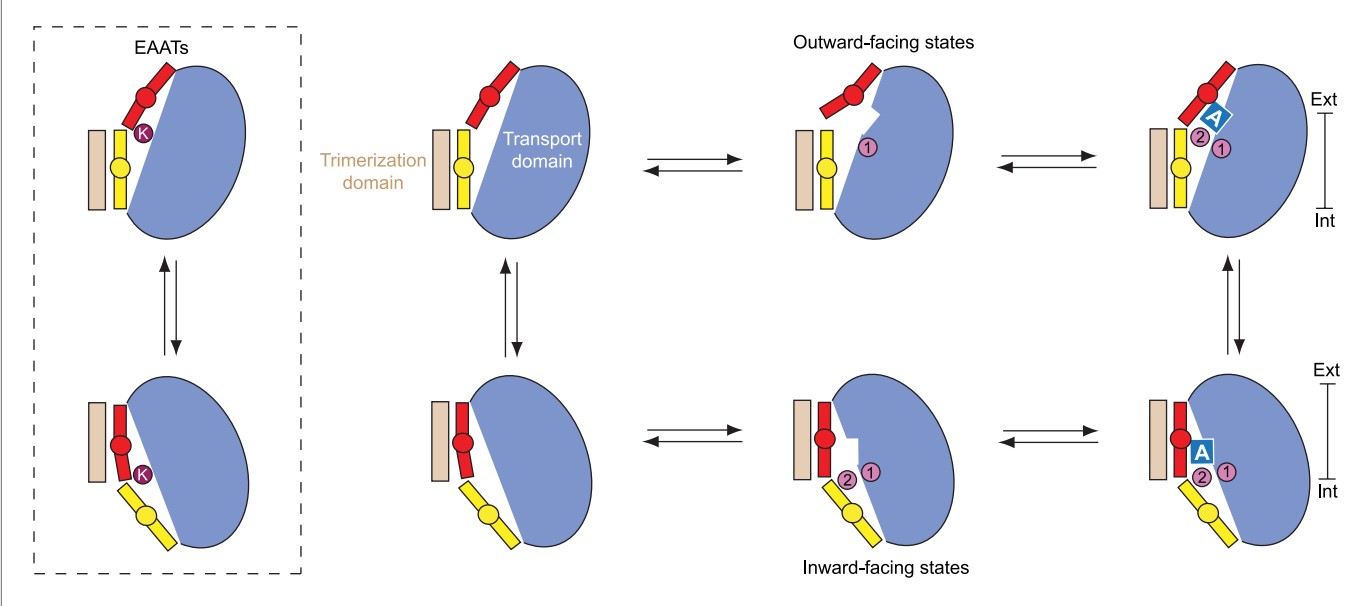

**Figure 10**. Proposed transport cycle for Glt$_{Ph}$ and EAATs. Ion binding to the Na1 site of the outward-facing apo transport domain triggers isomerization into bound-like conformation, formation of the L-asp and Na2 binding sites and HP2 opening, impeding translocation of the domain. Closure of HP2, coupled to L-asp and Na2 binding, allows translocation. After the release of the ligands into the cytoplasm by as yet an unknown gating mechanism, the domain is in a compact apo state, and returns to the extracellular side. Notably, binding of cations to the inward-facing state does not lead to a crystallographically observed gate opening that would impede translocation. However, Na$^+$ affinity in this state is only ~250 mM (**Reyes et al., 2013**), and it will remain largely unbound when facing the cytoplasm. Hence, uncoupled Na$^+$ transport should be limited. In EAATs, an open conformation of the gates might be more favored in the apo state, and K$^+$ binding at the Ct site might be required to stabilize translocation-competent conformation of the apo transport domain.

of the Na3 binding site; the potential role of the Ct site in binding K$^+$; and the gating mechanism in the inward-facing state will be necessary to verify and refine our proposed mechanisms.

## Materials and methods

### DNA constructs, mutagenesis, protein expression, and purification

R397A mutation was introduced by PCR into Glt$_{Ph}$ containing seven point mutations to histidine (**Yernool et al., 2004**), referred as wild type Glt$_{Ph}$ for brevity. Proteins were produced in *Escherichia coli* DH10b strain (Invitrogen, Inc., Grand Island, NY) as fusions with a thrombin cleavage site, and a metal-affinity octa-histidine at their carboxyl-terminus. Proteins were purified by nickel-affinity chromatography, digested with thrombin to remove the affinity tag, and purified by size exclusion chromatography (SEC) in appropriate buffers as described below. Protein concentrations were determined by measuring the absorbance at 280 nm using an extinction coefficient of 26,820 M$^{-1}$.cm$^{-1}$.

### Crystallization, and soaking experiments

#### Outward-facing state

Glt$_{Ph}$-R397A was purified by SEC in 10 mM HEPES/Tris, pH 7.4, 200 mM choline chloride and 7 mM *n*-decyl-α-D-maltopyranoside (Anatrace, inc., Maumee, OH). Crystallization experiments were setup using the hanging-drop vapor diffusion method, by mixing protein (~7 mg/ml) and well solutions (1:1 vol:vol), and incubated at 4°C. Na$^+$-bound Glt$_{Ph}$-R397A crystals were grown in 18–20% PEG 400, 0.1 M citric acid/Tris pH 4.5 and 0.4 M NaCl. Fully bound Glt$_{Ph}$-R397A crystals were obtained in the same crystallization conditions supplemented with 5 mM L-asp. The crystals were cryo-protected by soaking in the well solution supplemented with 10% glycerol and 7 mM *n*-decyl-α-D-maltopyranoside and frozen in liquid nitrogen. Apo crystals were grown in 18–20% PEG 400, 0.1 M citric acid/Tris pH 4.5, and 0.4 M choline chloride. Tl$^+$-bound Glt$_{Ph}$-R397A crystals were obtained by soaking Glt$_{Ph}$-R397A apo crystals in several changes of solutions containing 18–20% PEG 400, 0.1 M citric acid/Tris pH 4.5, 0.1 M TlNO$_3$, 7 mM *n*-decyl-α-D-maltopyranoside and 10% glycerol.

### Inward-facing state

$Glt_{Ph}$-K55C-A364C was purified by SEC in 10 mM HEPES/KOH, pH 7.4, 150 mM KCl, and 7 mM *n*-decyl-α-D-maltopyranoside. Cross-linking was carried out by mixing protein at ~3 mg/ml with a 10-fold molar excess of $HgCl_2$ for 20 min on ice. The samples were diluted ~10-fold with the SEC buffer and concentrated to remove partially the excess of $HgCl_2$. For crystallization, the protein samples at 2.75 mg/ml were supplemented with 20–50 mM $MgCl_2$ and 0.2 to 0.5 mM of a mixture of *E. coli* polar lipid extract and egg PC (3:1 wt:wt) (Avanti Polar Lipids, Inc., Alabaster, AL) and incubated on ice for 45 min. Crystallization was carried out using hanging drop diffusion methods in 96-well plates at 4°C. Initial crystallization conditions were identified using a replica of the crystallization screen MemGold (Molecular Dimensions., Altamonte Springs, FL), in which $Na^+$-containing compounds were replaced with $K^+$-containing compounds. The screen was prepared using the liquid handler Formulator (Formulatrix, Inc., Waltham, MA). $Glt_{Ph}^{in}$ crystals were grown in 14–20% PEG 400, and 0.1 M potassium citrate, pH 5.0 to 6.0. For $Tl^+$ soaking, crystals were washed in 20% PEG 400, 20 mM MES/Tris, pH 6.5, 20 mM $KNO_3$, 20 mM $MgNO_3$, and 10 mM *n*-decyl-α-D-maltopyranoside, and then incubated in 20% PEG 400, 20 mM MES/Tris, pH 6.5, 5 mM $MgNO_3$, 10 mM *n*-decyl-α-D-maltopyranoside, and 150 mM $TlNO_3$. In ion competition experiments, the soaking solution was further supplemented with either $NaNO_3$ or $KNO_3$. To obtain an alkali-free structure, crystals were soaked in solution containing 20% PEG 400, 20 mM MES/Tris, pH 6.5, 5 mM $MgCl_2$, and 10 mM *n*-decyl-α-D-maltopyranoside. Crystals were directly frozen in liquid nitrogen.

### Data collection and structure determination

Diffraction data were collected at the National Synchrotron Light Source beamlines X25 and X29 (Brookhaven National Laboratory). Data from crystals soaked in $Tl^+$ were collected at a wavelength of 0.97 Å. Data were processed using HKL2000 (*Otwinowski and Minor, 1997*), and further analyzed using the CCP4 program suite (*Collaborative Computational Project, 1994*). Anisotropy correction was performed as described previously (*Strong et al., 2006*). Briefly, resolution limits along the *a*, *b*, and *c* axes were determined using the UCLA–MBI Diffraction Anisotropy server (http://services.mbi.ucla.edu/anisoscale/) and applied as cutoffs to truncate the dataset obtained after processing of diffraction images. After scaling in HKL2000, structure factors were anisotropically scaled using PHASER (*McCoy et al., 2007*), and a negative B factor correction was applied to these structure factors using CAD. Initial phases were determined by molecular replacement with PHASER (*McCoy et al., 2007*), using the structure of $Glt_{Ph}$ either in the outward-facing state (PDB code 2NWX) or the inward-facing state (PDB code 3KBC) as the search model. Refinement was carried out by rounds of manual model building in COOT (*Emsley and Cowtan, 2004*) and refinement in REFMAC5 with TLS (*Winn et al., 2001*). With the exception of the analysis of the data from alkali-free $Glt_{Ph}^{in}$ crystals, where protomers in the trimer were clearly not identical, the electron density maps and the anomalous difference Fourier maps were three or sixfold averaged in real space. Strict non-crystallographic symmetry constrains were also applied during structural refinement in REFMAC5 when necessary. Structures of the transport domain were superimposed and R.M.S.D.s calculated using VMD software (*Humphrey et al., 1996*). All structural figures were prepared using Pymol (DeLano Scientific, LLC) (*DeLano, 2008*).

### Isothermal titration calorimetry (ITC)

ITC experiments were performed as described previously (*Reyes et al., 2013*). Briefly, $Glt_{Ph}$ mutant proteins were purified by SEC in 10 mM HEPES/Tris, pH 7.4, 200 mM choline chloride, 0.5 mM *n*-dodecyl-β-D-maltopyranoside and concentrated to 4 mg/ml. The protein was diluted to 40 μM in buffer containing 20 mM HEPES/Tris, pH 7.4, 200 mM choline chloride, 1 mM *n*-dodecyl-β-D-maltopyranoside and various NaCl concentrations. ITC experiments were performed using a small cell NANO ITC (TA instruments, Inc., New Castle, DE) at 25°C. Protein samples were placed into the instrument cell and titrated with L-asp solution prepared in the same buffer. The isotherms were analyzed using the NanoAnalyze software (TA instruments, Inc., New Castle, DE), and fitted to independent binding sites model.

### Fluorescence-based binding assays

Fluorescence-based binding assays were performed as described previously (*Reyes et al., 2013*). In brief, 100 μg/ml of protein in 20 mM HEPES/Tris, pH 7.4, 200 mM choline chloride, 0.4 mM *n*-dodecyl-β-D-maltopyranoside, 200 nM styryl fluorescent dye RH421 (Invitrogen, Inc., Grand Island, NY) were titrated with L-asp in the presence of various concentrations of NaCl at 25°C. Fluorescence experiments were carried out using a QuantaMaster (Photon International Technology, Inc., Edison, NJ). The RH421 dye was

excited at 532 nm, and the fluorescence was collected at 628 nm. Fluorescence emissions were measured after at least 1000 s equilibration. The data were analyzed using SigmaPlot12 (Systat software, Inc., San Jose, CA). Fractional fluorescence changes were corrected and normalized with respect to the dilution factors and maximal fluorescence changes, respectively. Corrected fluorescence changes were plotted as a function of ligand concentration and fitted to the Hill equation. Sodium activity was calculated as $\gamma \times [Na^+]$, where $\gamma$ is the activity coefficient. The activity coefficient is calculated with the Debye-Hückel equation as described (*Reyes et al., 2013*). All the experiments were performed at least in triplicate.

## Sequence analysis

Sodium:dicarboxylate symporter family sequences were harvested from PFAM database (PF00375) (*Finn et al., 2008*), parsed to remove incomplete sequences and sequences with over 70% identity and aligned in ClustalW (*Larkin et al., 2007*). The alignment was manually adjusted and the final dataset containing 463 aligned sequences was used to generate a consensus sequence using WebLogo (*Crooks et al., 2004*).

## Molecular modeling

To model the structures of the transport domains with HP2 in the bound conformation and the NMD motif in the apo conformation, and *vice versa*, we superimposed the structures of the fully bound and apo forms of the transport domains using TM6, HP1, and TM7. We then generated new coordinates files combining the coordinates of TM7, including the NMD motif, from the bound form and the coordinates for HP2 from the apo form or *vice versa*. In both of these models, we observed steric clashes between Met311 and residues in HP2. To construct a model of the intermediate state with an open tip of HP2, we superimposed the structure of $Na^+$-only bound $Glt_{Ph}$-R397A and the intermediate state (PDB accession code 3V8G) using TM6, HP1, and TM7. We then replaced HP2 in the structure of the intermediate with HP2 from $Na^+$-only bound $Glt_{Ph}$-R397A. We moved slightly the side chain of Lys55 that was involved in a minor steric clash with the HP2 tip. We observed no major steric clashes in the resulting model.

## Acknowledgements

Use of the National Synchrotron Light Source, Brookhaven National Laboratory, was supported by the US Department of Energy, Office of Science, Office of Basic Energy Sciences, under Contract No. DE-AC02-98CH10886. We thank Howard H Robinson, Annie Héroux for access to beamlines X29 and X25 at the National Synchrotron Light Source, and the staff for their assistance. We thank Deena Oren (Structural Biology Resource Center, The Rockefeller University) for her assistance with the liquid handling robot. We also thank Michael Sawaya (Institute for Genomics and Proteomics, University of California, Los Angeles, USA) for providing the programs and scripts for performing the anisotropy correction. We thank Irene Kiburu for the advice on calculation of R.M.S.D.s in VMD. We thank Alessio Accardi, Nurunisa Akyuz, Crina Nimegean and Nicolas Reyes for helpful comments on the manuscript.

Contributions

GV, SO, and OB designed the experiments and analyzed the data. GV and SO conducted the experiments, and collected X-ray diffraction data. RS performed preliminary ion-binding experiments for $Glt_{Ph}{}^{in}$-M311A. GV, OB, and SO wrote the manuscript.

## Additional information

### Funding

| Funder | Grant reference number | Author |
| --- | --- | --- |
| National institutes of Health | NS064357 | Olga Boudker |

The funder had no role in study design, data collection and interpretation, or the decision to submit the work for publication.

### Author contributions

GV, SO, Conception and design, Acquisition of data, Analysis and interpretation of data, Drafting or revising the article; RNS, Acquisition of data, Analysis and interpretation of data; OB, Conception and design, Analysis and interpretation of data, Drafting or revising the article

# Additional files

## Major datasets

The following datasets were generated:

| Author(s) | Year | Dataset title | Dataset ID and/or URL | Database, license, and accessibility information |
|---|---|---|---|---|
| Verdon G, Boudker O | 2014 | Closed, apo inward-facing state of the glutamate transporter homologue GltPh | http://www.pdb.org/pdb/explore/explore.do?structureId=4p19 | Publicly available at RCSB Protein Data Bank. |
| Verdon G, Boudker O | 2014 | Thallium-bound inward-facing state of the glutamate transporter homologue GltPh | http://www.pdb.org/pdb/explore/explore.do?structureId=4p1a | Publicly available at RCSB Protein Data Bank. |
| Verdon G, Boudker O | 2014 | Apo inward-facing state of the glutamate transporter homologue GltPh in alkali-free conditions | http://www.pdb.org/pdb/explore/explore.do?structureId=4p3j | Publicly available at RCSB Protein Data Bank. |
| Verdon G, Boudker O | 2014 | Tl+-bound inward-facing state (bound conformation) of the glutamate transporter homologue GltPh | http://www.pdb.org/pdb/explore/explore.do?structureId=4p6h | Publicly available at RCSB Protein Data Bank. |
| Boudker O, Oh S | 2014 | Crystal structure of GltPh R397A in apo | http://www.pdb.org/pdb/explore/explore.do?structureId=4oye | Publicly available at RCSB Protein Data Bank. |
| Boudker O, Oh S, Verdon G, Serio R | 2014 | Crystal structure of GLTPH R397A IN Sodium-bound state | http://www.pdb.org/pdb/explore/explore.do?structureId=4oyf | Publicly available at RCSB Protein Data Bank. |
| Boudker O, Oh S | 2014 | Crystal structure of GltPh R397A in complex with Na+ and L-asp | http://www.pdb.org/pdb/explore/explore.do?structureId=4oyg | Publicly available at RCSB Protein Data Bank. |

The following previously published datasets were used:

| Author(s) | Year | Dataset title | Dataset ID and/or URL | Database, license, and accessibility information |
|---|---|---|---|---|
| Boudker O, Ryan RM, Yernool D, Shimamoto K, Gouaux E | 2007 | Crystal structure of GltPh in complex with L-aspartate and sodium ions | http://www.pdb.org/pdb/explore/explore.do?structureId=2nwx | Publicly available at RCSB Protein Data Bank. |
| Reyes N, Ginter C, Boudker O | 2009 | Crystal structure of GltPh K55C-A364C mutant crosslinked with divalent mercury | http://www.pdb.org/pdb/explore/explore.do?structureId=3kbc | Publicly available at RCSB Protein Data Bank. |
| Verdon G, Boudker O | 2012 | Crystal structure of an asymmetric trimer of a glutamate transporter homologue (GltPh) | http://www.pdb.org/pdb/explore/explore.do?structureId=3v8g | Publicly available at RCSB Protein Data Bank. |

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
