## [Decision Letter]

Thank you for sending your work entitled “Coupled ion binding and structural transitions along the transport cycle of glutamate transporters” for consideration at *eLife*. Your article has been very favorably evaluated by a Senior editor, John Kuriyan, a Reviewing editor, and 3 reviewers, one of whom, Lucy Forrest, has agreed to reveal her identity.

The Reviewing editor and the reviewers discussed their comments before we reached this decision, and the Senior editor has assembled the following comments to help you prepare a revised submission.

Boudker and colleagues present a manuscript describing structures and binding data for the prokaryotic asparate transporter GltPh, an important reference system for the neuronal glutamate transporters of the EAAT family. The manuscript contains an array of important data, including substrate-free states, and sodium-only states, that together provide crucial insights into ion-substrate coupling and transporter conformational dynamics. The several crystal structures for ion-bound apo state allow for the first time a dissection of the exact mechanism of coupling in this system. The findings are in line with previous work and the proposed mechanisms of translocation of the apo states and the Na coupling makes perfect sense. The discussion on a tentative location of K+ binding site, while offering a plausible explanation (and a novel one) remains a little speculative. However, the authors are careful in the interpretation of the K sites and these sites might be the ones used in EAATs. Nevertheless, this is novel and rigorous study of a high caliber! One of the reviewers describes the paper as “a great manuscript”.

The following comments are collated from the individual reviews. The authors should pay attention to them when submitting a revised manuscript, and provide an itemized response. The revised manuscript will be dealt with by the editors, and will not be sent out for further external review, so you should expect a quick final decision.

Specific comments:

*Reviewer 1*:

1) Introduction section; not clear what “reflect weaker coupling” refers to. The slope only tells something about how many Na could be involved in the conformational changes, not anything about coupling. Maybe it would be good to clearly state what the measurement really mean. A Hill coefficient of 1 doesn't mean that only 1 ion binds. Is this the same case here?

2) The off rate of aspartate is dependent on all conformational changes in the cycle and the coupling could be very indirect by Na affecting another step in the cycle. This should be stated.

3) Not clear what is meant by “trigger isomerization from apo... to bound ... conformation”; Isomerization of what?

4) It is also not clear why the isomerization would make the Na affinity low. Previous molecular modelling has proposed that Na binds with low affinity to the GltPh sites (without any isomerization needed to explain this low affinity).

5) Are the two Tl^+^ sites occupied at the same time or in different crystals?

6) Figure 10. It is not clear from the model figure how the substrate and ions come of the transporter in the inward state?

7) In the Introduction, it is stated that Na1 binding form the Na2 and asp binding sites. However, this is confusing, since also Na1 binding opens the Hp2 gate which should prevent Na2 binding. Please state carefully what Na1 does.

8) Figure 6 is not very legible.

9) Previous experiments and modeling have suggested that the apo state returns by adopting a similar conformation as the fully loaded transporter. The present crystals fully support this.

*Reviewer 2*:

1) General remark. R/A mutation used in this paper may affect overall electrostatic of the pocket and thus bias packing of ions. Some cautious remarks should be put in the manuscript.

2) While describing replacement of M311 with Leu the authors states: “similarly bulky” for L311 replacement. I am unsure if it is exactly correct statement. Furthermore, M311 was assigned a variety of roles in the past including tentative participation in stabilization of Tl^+^ in the binding pocket due to high polarizability. Any comments/discussions to that end would be beneficial. Tl^+^ marking of Na1/Na2 site remains a little bit of a mystery. Thus, it is not only volume but also chemistry of a side-chain that may be important.

3) The hypothesis about ion “priming” the site for substrate acceptance is interesting. Zhao et al. (PLoS Comput Biol 9(10): e1003296) described recently a fairly similar mechanism for MhP1 where transporter appears to get “primed” for substrate by binding of an ion stabilizing one of the states. Some discussion on this seemingly general mechanism, which may exist in other systems may benefit this manuscript. It is entirely up to the authors, but again it may represent a general mechanism of allosteric coupling during secondary transport.

4) Furthermore, the idea of shuttling of a “compact” state of this transporter is interesting and well supported by two side-by-side publications in Nat Stuct Biology, but still some additional discussion on possible energetics of this process would be advantageous! The large protein has to move a great length across the membrane plain.

*Reviewer 3*:

1) My only substantive query is why several of the structures have not been deposited in the PDB. These are all structures at >4 Angstrom resolution, but I'm not sure that is sufficient reason; this may come down to journal policy.

2) I didn't follow the arguments regarding the evidence for priming of the transporter by Na1. For example, the Na2 site is distorted in GltPh-R397A+Na, so its structure is not really similar to the bound state. Maybe it would help to separate out the figures of the outward-open R397A+Na from those of the inward-facing Tl-bound. Or perhaps the phrasing just needs a little reworking.

3) In the Introduction section I was confused by these statements: “Met311 in the NMD motif is the only residue that is shared between the Na1 site and the substrate and Na2 sites and also undergoes a conformational change upon ligand binding.” Which are the reference states in each case? Are the authors only comparing the trimerization domains?

4) I would welcome quantitation of the similarity between domains in the different states, using RMSDs of the backbone and of all atoms. It would put the differences in context relative to the crystallographic resolution.

5) In Figure 4—figure supplement 1, it is not yet explained what is meant by the Ct site, so the labeling is confusing.

6) It would be good to label/show S92 and T93 in TM3 in Figure 3 or Figure 3–supplement, as they are referenced in the text.

7) The phrase “in the context of unconstrained GltPh and GltPh-in” is not immediately clear.

8) The legend of Figure 6 should clarify which structure is in colors. In that figure, I found it difficult to keep track of which structure was which.

9) Which figure is referred to for the Tl data? Could the authors comment on why Na2 is bound in GltPh-in in the absence of substrate? How does this relate to the point about coupling requiring?

10) I would say that since Figure 7 does not illustrate the progression of the transport domain beyond the intermediate structure, the authors should rephrase to “because the tip would presumably clash with TM5”

11) Figure 9 legend: I think that the word dynamic should be removed from the legend title; the figure clearly shows changes between two structures under two conditions, but dynamics aren't clearly demonstrated by the figure.

12) PDB identifiers will need to be added to the tables.

---

## [Author Response]

Reviewer 1:

*1) Introduction section; not clear what “reflect weaker coupling” refers to. The slope only tells something about how many Na could be involved in the conformational changes*, *not anything about coupling. Maybe it would be good to clearly state what the measurement really mean. A Hill coefficient of 1 doesn't mean that only 1 ion binds. Is this the same case here?*

We have rephrased this sentence. It now reads: “reduced binding cooperativity between the substrate and Na^+^ ions”. Indeed, we do not imply that fewer Na^+^ ions bind to the transporter, only that the extent of cooperativity between binding of the ions and the substrate is diminished.

*2) The off rate of aspartate is dependent on all conformational changes in the cycle and the coupling could be very indirect by Na affecting another step in the cycle. This should be stated*.

This could be indeed the case in the wild-type transporter. However, we note that the loss of cooperativity is also observed in the transporter that is cross-linked in the inward facing state, suggesting that binding/dissociation in this isolated state is affected. We hope that using “binding cooperativity” rather than “coupling” in the revised manuscript eliminates the ambiguity.

3) Not clear what is meant by “trigger isomerization from apo... to bound ... conformation”: Isomerization of what?

Here, we refer to the isomerization of the transport domain. We have made the corresponding correction in the text.

*4) It is also not clear why the isomerization would make the Na affinity low*. *Previous molecular modelling has proposed that Na binds with low affinity to the GltPh sites (without any isomerization needed to explain this low affinity).*

Computational studies appear to predict binding free energies for Na1 ranging from – 7 kcal/mol (Heinzelmann et al., 2013) to -12.5 kcal/mol (Larsson et al., 2010). We have restated this more clearly in the revised text and included the references.

*5) Are the two Tl*^*+*^
*sites occupied at the same time or in different crystals?*

To clarify, we soaked crystals of the apo crystals of the inward facing GltPhin in Tl containing buffers. Although the same soaking conditions were used, we obtained two types of crystals. In one, Tl^+^ ions incorporated at the canonical Na1 and Na2 sites, and the protein was in a bound -like conformation. In the other, Tl^+^ ions incorporated at the two new Ct and Na2’ sites in the apo conformation. Both Ct and Na2’ sites were occupied in all three protomers. To determine whether the two sites were occupied at the same time in each molecule in the crystal, we would have to refine the occupancies of the Tl sites. Our occupancy estimates are 40-80%. However, the resolution of the data is not sufficient to estimate reliably these occupancies. Hence, we think that the sites are very likely occupied at the same time, but cannot state this with certainty. Therefore, we did not include such discussion in the text.

*6)*
Figure 10*. It is not clear from the model figure how the substrate and ions come of the transporter in the inward state?*

At present, the release/gating mechanism is still not known. We have added a comment in the legend of Figure 10.

*7) In the Introduction, it is stated that Na1 binding form the Na2 and asp binding sites. However, this is confusing, since also Na1 binding opens the Hp2 gate which should prevent Na2 binding. Please state carefully what Na1 does*.

We have restated this more clearly. In brief, Na^+^ binding to the Na1 site is coupled through movements of Met311 to the structural transitions in HP2. While HP2 is collapsed in the apo form of the transporter and occupies the substrate binding and Na2 sites, in the Na^+^-bound form it frees these sites and assumes conformations that are more similar to the fully bound form. We have clarified this point in the text.

*8)*
Figure 6
*is not very legible*.

We have increased the size of the panel and the font size for the residue numbers.

*9) Previous experiments and modeling have suggested that the apo state returns by adopting a similar conformation as the fully loaded transporter. The present crystals fully support this*.

We have included in the discussion a reference to Focke et al. (J. Neuroscience 2011), who have suggested that the apo form would be as compact and closed as the fully bound form.

Reviewer 2:

*1) General remark. R/A mutation used in this paper may affect overall electrostatic of the pocket and thus bias packing of ions. Some cautious remarks should be put in the manuscript*.

We have added the following statement in the Introduction section: “These results suggest that R397A is suitable to capture the apo and ions-only bound outward-facing states for their structural characterization. However, removal of Arg397 may affect local electrostatics, potentially altering ion binding; thus these studies should be interpreted with caution.”

*2) While describing replacement of M311 with Leu the authors states: “similarly bulky” for L311 replacement. I am unsure if it is exactly correct statement. Furthermore, M311 was assigned a variety of roles in the past including tentative participation in stabilization of Tl*^*+*^
*in the binding pocket due to high polarizability. Any comments/discussions to that end would be beneficial. Tl*^*+*^
*marking of Na1/Na2 site remains a little bit of a mystery*. *Thus, it is not only volume but also chemistry of a side-chain that may be important*.

We have replaced “similarly bulky” by “another bulky”. We agree with the reviewer that the role of the methionine residue side chain in Tl^+^ and/or Na^+^ binding remains poorly understood. We also note that from sequence analysis, it appears that Na^+^-coupled members of the family have Met at this position. In contrast, the members that are more likely to be proton-coupled (those, for example, that lack Asp405, one amino acid whose side chain is clearly involved in Na^+^-binding) have Leu at this position. While our data do not allow us to explain the role that the other group of Met311 may play in Na^+^ binding, we made a relevant comment in the revised manuscript.

*3) The hypothesis about ion “priming” the site for substrate acceptance is interesting. Zhao et al. (PLoS Comput Biol 9(10): e1003296) described recently a fairly similar mechanism for MhP1 where transporter appears to get “primed” for substrate by binding of an ion stabilizing one of the states. Some discussion on this seemingly general mechanism, which may exist in other systems may benefit this manuscript. It is entirely up to the authors, but again it may represent a general mechanism of allosteric coupling during secondary transport*.

We feel that the “priming” role for the Na^+^ ions in Mhp transporters proposed by Zhao and Noskov is distinct from their role in Glt_Ph_. In Mhp, the proposed key role of Na^+^ ions is to stabilize outward facing state compared to the inward facing state. In Glt_Ph_, we propose that the role of Na^+^ ions is local: structurally organizing the substrate-binding site within either the outward-facing or the inward-facing state. Hence, we feel that inclusion of the reference to Zhao and Noskov’s work may bring confusion. We have however added references to the crystal structures of Na^+^-only bound Mhp and LeuT transporters that, like Glt_Ph_, show open structures.

*4) Furthermore, the idea of shuttling of a “compact” state of this transporter is interesting and well supported by two side-by-side publications in Nat Stuct Biology, but still some additional discussion on possible energetics of this process would be advantageous! The large protein has to move a great length across the membrane plain*.

We agree with the reviewer that it is fascinating that such large conformational changes can occur readily in membranes. We note also that large and rapid transitions somewhat reminiscent of Glt_Ph_ have recently been proposed for the Na^+^/H^+^ exchanger. While the dynamics is an important question for us, the current manuscript does not add significantly to the topic, other than to suggest that the apo transport domain may move in a manner overall similar to that of the fully loaded protein. We feel that extending the discussion on energetics here would not be appropriate.

Reviewer 3:

*1) My only substantive query is why several of the structures have not been deposited in the PDB. These are all structures at >4 Angstrom resolution, but I'm not sure that is sufficient reason; this may come down to journal policy*.

We have also deposited the structure of Tl-bound GltPhin in the bound-like conformation. Regarding the other structures, we think that the structures do not provide additional structural information. They were used only to evaluate monitor the anomalous signal of Tl^+^ ions at the Ct and Na2’ sites.

*2) I didn't follow the arguments regarding the evidence for priming of the transporter by Na1. For example, the Na2 site is distorted in GltPh-R397A+Na, so its structure is not really similar to the bound state. Maybe it would help to separate out the figures of the outward-open R397A+Na from those of the inward-facing Tl-bound. Or perhaps the phrasing just needs a little reworking*.

We have improved this paragraph. The key clarification is: “The obtained outward- and inward-facing structures pictured the transport domains in conformations overall similar to those observed in the fully bound transporter: straightened TM3, Met311 pointing towards the binding sites, extended helix in HP2a and HP2 tip raised out of the substrate binding site (Figure 6).” The remainder of the paragraph better separates the discussion on the inward and outward facings states to avoid confusion. Finally, we added a figure (Figure 3—figure supplement 2), which shows RMSDs per residue between the transport domains of the various states presented. From this figure, it is clear that Na^+^-binding triggers most of the structural changes that are associated with loading of the transport domain.

*3) In the Introduction*
*section I*
*was confused by these statements: “Met311 in the NMD motif is the only residue that is shared between the Na1 site and the substrate and Na2 sites and also undergoes a conformational change upon ligand binding*.*” Which are the reference states in each case? Are the authors only comparing the trimerization domains?*

We wanted to point out that the positions and orientations of the transport domains relative to the trimerization domains are the same in the apo and fully bound forms. The conformations of the transport domains themselves are significantly different. We have rephrased the sentence.

*4) I would welcome quantitation of the similarity between domains in the different states, using RMSDs of the backbone and of all atoms. It would put the differences in context relative to the crystallographic resolution*.

We have generated new figures, including Figure 3—figure supplement 2 and Figure 9, where we show RMSDs for main chain atoms of the transport domains calculated between equivalent residues in different structural states. These figures clarify the extent of the structural changes and similarities between the various states.

*5) In*
Figure 4—figure supplement 1*, it is not yet explained what is meant by the Ct site, so the labeling is confusing*.

We have moved the figure and the corresponding discussion to the section on Ct site so that Figure 4—figure supplement 1 has become Figure 8—figure supplement 1.

*6) It would be good to label/show S92 and T93 in TM3 in*
Figure 3
*or*
*Figure 3–supplement**, as they are referenced in the text*.

We have added the labels in Figure 3 and Figure 3—figure supplement 1

*7) The phrase “in the context of unconstrained GltPh and GltPh-in” is not immediately clear*.

We have clarified this: “unconstrained wild-type GltPh and inward cross-linked Glt_Ph_^in^”

*8) The legend of*
Figure 6
*should clarify which structure is in colors. In that figure, I found it difficult to keep track of which structure was which*.

We have clarified the legend. We have also included the substrate in the figure.

9) Which figure is referred to for the Tl data? Could the authors comment on why Na2 is bound in GltPh-in in the absence of substrate? How does this relate to the point about coupling requiring?

The appropriate figure is Figure 6. We have referenced it in the text. We have also added the discussion on coupling. In brief, it is possible that the closed Tl^+^-bound inward-facing state observed in crystallo reflects the closed Na^+^-bound inward -facing state of the transporter. This does not violate the principle of coupled symport because the affinity for Na^+^ in this state is very low (Kd∼250 mM) and in the cytoplasm where the Na^+^ concentration is expected to be around ∼ 10 mM, so this state is not expected to be significantly populated.

*10) I would say that since*
Figure 7
*does not illustrate the progression of the transport domain beyond the intermediate structure, the authors should rephrase to “because the tip would presumably clash with TM5”*.

We have rephrased this part of the sentence as follows: “further progression of the transport domain to the inward-facing position could be impeded because the tip is likely to clash with TM5 in the trimerization domain”.

*11)*
Figure 9
*legend: I think that the word dynamic should be removed from the legend title; the figure clearly shows changes between two structures under two conditions, but dynamics aren't clearly demonstrated by the figure*.

We have changed the title to “Movements of the HP1-TM7a structural module”

*12) PDB identifiers will need to be added to the tables*.

PDB codes have been added to the tables.